# Revealing Weaknesses in Text Watermarking Through Self-Information Rewrite Attacks

Yixin Cheng [1 2]   Hongcheng Guo [3]   Yangming Li [4]   Leonid Sigal [1 2 5 6]

## Abstract

Text watermarking aims to subtly embeds statistical signals into text by controlling the Large Language Model (LLM)'s sampling process, enabling watermark detectors to verify that the output was generated by the specified model. The robustness of these watermarking algorithms has become a key factor in evaluating their effectiveness. Current text watermarking algorithms embed watermarks in high-entropy tokens to ensure text quality. In this paper, we reveal that this seemingly benign design can be exploited by attackers, posing a significant risk to the robustness of the watermark. We introduce a generic efficient paraphrasing attack, the Self-Information Rewrite Attack (SIRA), which leverages the vulnerability by calculating the self-information of each token to identify potential pattern tokens and perform targeted attack. Our work exposes a widely prevalent vulnerability in current watermarking algorithms. The experimental results show SIRA achieves nearly 100% attack success rates on seven recent watermarking methods with only $0.88 per million tokens cost. Our approach does not require any access to the watermark algorithms or the watermarked LLM and can seamlessly transfer to any LLM as the attack model even mobile-level models. Our findings highlight the urgent need for more robust watermarking. The source code is available at SIRA.

## 1. Introduction

Large language models (LLMs), exemplified by ChatGPT (OpenAI, 2024) and Claude (Anthropic, 2024), have demonstrated remarkable capabilities in generating coherent, human-like text. However, while these advances significantly expand AI's potential in content creation, they have concurrently heightened concerns regarding their misuse (Deshpande et al., 2023; Wang et al., 2024), including the spread of misinformation (Monteith et al., 2024) and threats to academic integrity (Stokel-Walker, 2022).

To mitigate risks associated with LLM-generated content, text watermarking has emerged as a promising countermeasure (Kirchenbauer et al., 2023; Aaronson & Kirchner, 2022). This technique subtly alters the LLM's generation process to embed imperceptible patterns in the output text, the pattern is invisible to human readers and can be reliably detected using specialized algorithms. This generate-detect framework enables the differentiation between AI-generated and human-authored content and allows tracking of the text back to the specific LLM that generates the text (Li et al., 2024). Consequently, this mechanism promotes accountability and helps mitigate LLM misuse, providing a reliable means to ensure transparency and integrity in AI-generated content.

Recent studies have demonstrated that watermarking techniques exhibit significant robustness against simple manipulations, including word deletions (Welbl et al., 2020) and emoji attacks (Kirchenbauer et al., 2023). However, traditional NLP attack strategies, such as word deletion and insertion, are increasingly insufficient for thoroughly evaluating the robustness of advanced watermarking algorithms. As LLMs continue to advance, there is a growing need for more sophisticated testing methodologies that account for complex manipulation tactics, ensuring that watermarking techniques remain resilient against emerging threats.

To provide a more rigorous evaluation of watermarking robustness, paraphrasing attacks have been proposed. Despite their potential, these approaches face several limitations. First, current paraphrasing attacks rely on a naive and brute-force approach, where they simply instruct LLMs to rewrite watermarked text. This process is both inefficient and inconsistent in its results. The modifications to text are untargeted and random, dictated by the LLM, often leaving portions of the watermark intact causing the attack to fail. For newer and more robust watermarking algorithms like SIR (Liu

[1]University of British Columbia [2]Vector Institute for AI [3]Fudan University [4]University of Cambridge [5]Canada CIFAR AI Chair [6]NSERC CRC Chair. Correspondence to: Yixin Cheng <yixinch@cs.ubc.ca>.

*Proceedings of the $42^{st}$ International Conference on Machine Learning*, Vancouver, Canada. PMLR 267, 2025. Copyright 2025 by the author(s).

et al., 2024), these methods already fail to deliver effective attacks, making them unsuitable as robustness evaluation methods for future research. Moreover, changing the words that do not embed the watermark may cause a change in semantics, or lead to decline in text quality. Second, current methods require significant hardware resources and costs, as they often rely on large-scale LLMs to achieve notable attack performance which could be a barrier for future study. Thirdly, they are non-transferable, as the reliance on specifically fine-tuned LLM (Krishna et al., 2024) prevents them from effectively leveraging the capabilities of more powerful, rapidly emerging language models for attacks.

To address these challenges, we propose a new paraphrasing attack named **SIRA** (Self-Information Rewrite Attack). Our approach not only introduces a more effective paraphrasing strategy but also reveals a fundamental vulnerability in current watermark algorithms. Specifically, watermarking techniques aim to be imperceptible to users while maintaining text quality and semantics intact, which necessitates embedding patterns in high-entropy tokens (Kirchenbauer et al., 2023; Liu et al., 2023). These high-entropy tokens also exhibit high self-information within the given text context. Leveraging this otherwise harmless watermark feature, SIRA could identify potential "green list" token candidates within watermarked text without any prior knowledge. By masking potential green tokens, we can transform the untargeted paraphrasing into a targeted fill-in-the-blank task, achieving a stronger and more efficient attack.

In summary, we make the following contributions:

- We formalize and thoroughly investigate the threat model of LLM watermark black-box paraphrasing, distinguishing it from other watermark attacks that rely on probing-based modeling or online attack methods.

- We reveal a widely existing vulnerability in watermark algorithms and propose the first, to our knowledge, targeted paraphrasing attack. This attack is easy to implement and transferable, making it well-suited for future robustness evaluations.

- We comprehensively study paraphrasing attacks on recent watermark algorithms. For newly proposed watermarking algorithms, we show that existing paraphrasing attacks are no longer sufficient to verify robustness.

## 2. Related Work

**Self-information.** Self-information, also known as surprisal, is a fundamental concept in Information Theory, first introduced by Claude Shannon in his seminal work (Shannon, 1948). Shannon employed self-information as the principal metric to quantify the information content associated with the occurrence of specific events, effectively linking the rarity of an event to the amount of information it communicates.

In the realm of Natural Language Processing (NLP), self-information plays a crucial role in the analysis and modeling of language. It aids in deciphering language patterns, particularly in evaluating the entropy and predictability of tokens within sequences. The concept is particularly useful for quantifying the informativeness or surprise of a token in a given linguistic context. Language models predict the probability of a subsequent token in a sequence using the preceding context $P(t_k \mid t_1, t_2, \ldots, t_{k-1})$. The self-information of the token in this context is computed as follows:

$$I(t_k \mid t_1, t_2, \ldots, t_{k-1}) = -\log_b(P(t_k \mid t_1, t_2, \ldots, t_{k-1}))$$

where $I(t_k \mid t_1, t_2, \ldots, t_{k-1})$ denotes the self-information of token $t_k$ given the context of previous tokens, $P(t_k \mid t_1, t_2, \ldots, t_{k-1})$ is the probability of token $t_k$ occurring after the preceding sequence of tokens, and $b$ represents the base of the logarithm.

**LLM Watermark.** Watermarking techniques for large language models are designed to embed identifiable patterns in model outputs, allowing for the traceability of generated text back to its originating source. These watermarks serve as an essential tool for ensuring accountability and ownership, particularly in scenarios where identifying the specific model or version that produced the content is crucial. LLM watermark methods can be broadly classified into two primary categories: the KGW Family and the Christ Family. Each family employs distinct mechanisms that are integral to the internal workings of LLMs. The *KGW Family* (Kirchenbauer et al., 2023; Liu et al., 2023; Zhao et al., 2023; Wu et al., 2024; Lu et al., 2024) focuses on modifying the logits—the raw output probabilities produced by the model—before they are transformed into text. This approach involves selectively adding bias to certain tokens, referred to as "green list" tokens, which influences the model to favor these tokens, thus embedding a statistical signature in the output. Post text generation, a statistical metric based on the proportion of these "green" tokens is computed. A predetermined threshold enables differentiation between watermarked and non-watermarked text.

Conversely, the *Christ Family* (Aaronson & Kirchner, 2022; Christ et al., 2024; Kuditipudi et al., 2023) modifies the sampling process during the generation phase itself. Rather than altering the logits, this family intervenes directly in the token selection during decoding. Techniques such as top-k sampling, temperature adjustment, or nucleus sampling are adapted to ensure preferential selection of watermarked tokens. This method provides more direct control over the generation process, embedding watermarks that are resilient against post-processing attacks, such as paraphrasing.

**Watermark Removal Attacks.** The robustness of a watermarking algorithm is crucial, as it determines the effectiveness of the watermark under various real-world conditions, particularly in adversarial settings. Attacks against watermark algorithms, commonly referred to as watermark tampering attacks, can be broadly categorized into two types:

*Text manipulations*: These attacks involve traditional NLP techniques to straightforward text manipulations, such as word deletion (Welbl et al., 2020), substitution (Yu et al., 2010), or insertion (Kirchenbauer et al., 2023). By altering the word-level structure of the text, these methods attempt to distort or eliminate the watermark. These techniques disrupt the statistical pattern of the watermark by adding, removing, or replacing words. Kirchenbauer et al. (2023) propose emoji attack and copy-paste attack which insert emoji/human written text in the generated text to avoid detection. These methods are considered variants of text manipulations, however, they are easily thwarted by detectors equipped with content filters and often alter the semantics of the generated text which makes them inappropriate for real-world use.

*Informed watermark attack*: Such attacks rely on the adversary having specific knowledge or access to the watermarking system, such as the ability to probe the watermark model. One typical attack is the watermark-stealing attack (Jovanović et al., 2024; Wu & Chandrasekaran, 2024). These methods aim to reverse-engineer or approximate the embedded watermark by probing watermark algorithms with queries. Attackers construct the watermark distribution of the target model and estimate whether each token complies with the watermark rules based on the context by issuing a large number of pre-designed prefix queries. However, these methods assume that the adversary has unlimited access to the targeted watermarked LLM model. Additionally, this method becomes ineffective when watermarks employ dynamic strategies (e.g., using a set of hash keys). Another notable recent attack is the random walk attack (Zhang et al., 2023); this method iteratively perturbs the watermarked text using multiple models, relying on the detector's feedback as the termination condition. While it effectively removes watermarks, its iterative nature introduces significant computational and time overhead. Another major limitation of this approach is it does not guarantee any semantic preservation.

The above methods share a strong assumption: the attacker has access to the watermarked LLM, either with or without the watermark detector. However, this assumption is rarely met in real-world adversarial scenarios especially the access to the watermark detector. The aforementioned methods also suffer from significant overhead, both in terms of computational resource demands and execution time. As a result, these methods are generally excluded from robustness evaluations of watermarking algorithms Kirchenbauer et al. (2023); Liu et al. (2024); Aaronson & Kirchner (2022); Zhao et al. (2023). *Due to the different threat models and much stronger assumptions, we do not include them in this paper.*

*Model-based paraphrasing*: A more common and advance form of attack involves using another LLM to paraphrase the watermarked content. This method differs from watermark-stealing attacks by operating in a strict black box setting, where the attacker's knowledge is limited to the watermarked text. Krishna et al. (2024) propose DIPPER, a paraphrase generation model developed by fine-tuning T5-XXL (Raffel et al., 2020b) on an aligned paragraph dataset. This model has been widely adopted in recent watermark research (Zhao et al., 2023; Liu et al., 2024; Kuditipudi et al., 2023) to evaluate the robustness of watermarking algorithms. GPT Paraphraser is another model-based method, which utilizes the GPT model as a paraphrase. Previous approaches in this category derive their effectiveness primarily from two mechanisms: (1) paraphrased text especially new generated portion dilutes the strength of the original watermark by introducing new linguistic structures, and (2) lexical variations introduced during paraphrasing could incidentally alter a subset of green tokens. However, such changes that are controlled by an LLM, thus occurring *by chance*, leads to untargeted paraphrasing that is neither sufficient nor effective. In contrast, SIRA is a targeted approach that selectively replaces potential green tokens in the watermarked text, creating a "neutral" template for rewriting. It transforms the paraphrasing task into a fill-in-the-blank problem.

Our proposed SIRA falls under the category of model-based paraphrasing attacks and offers several key advantages: (1) **High Attack Effectiveness**: SIRA achieves near 100% success rates across seven tested watermarking algorithms; (2) **Lightweight and Highly Transferable**: Unlike DIPPER, which depends on a specific fine-tuned model, SIRA operates effectively on mobile-level 3B models and can seamlessly adapt to future powerful model; (3) **Minimal Assumptions**: It requires no access to the watermarked LLM or its detector, enabling robust performance in black-box scenarios; (4) **Low Cost and Plug-and-Play**: SIRA operates with minimal computational overhead (costing only$0.88 per Million tokens). Unlike watermark stealing requiring large samples of watermarked text to initial one piece text attack, SIRA is ready to use out of the box. These advantages position SIRA as a promising tool for evaluating the robustness of LLM watermarking in future research.

## 3. Methods

In this section, we detail SIRA attack formulation and implementation. First, we lay out the problem setting in Sec-

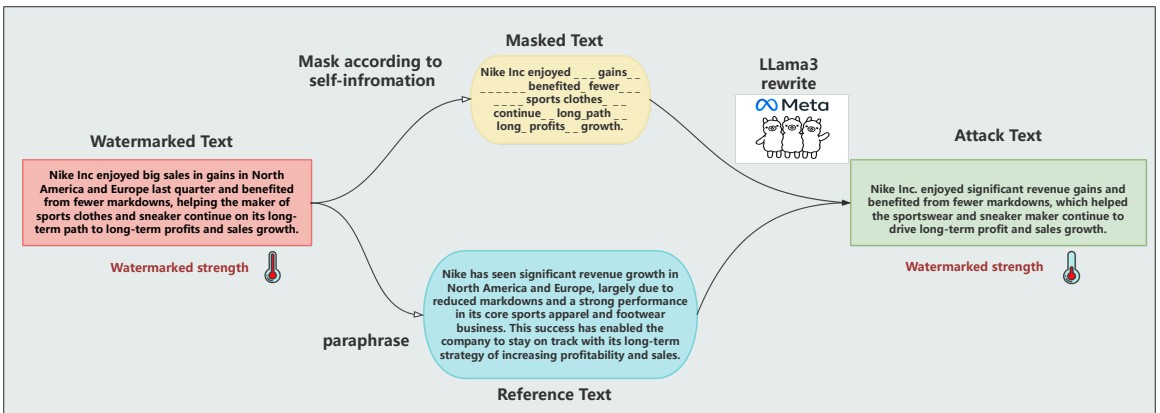

*Figure 1.* **SIRA pipeline consisting to two steps.** First, the attack generates a masked text based on self-information. If the self-information of a specific part above a pre-set threshold, that portion of the text is masked and replaced with a placeholder. Simultaneously, a reference text is generated by asking the LLM to paraphrase. In the second step, the LLM is prompted to complete the masked text while incorporating all the information from the reference text.

tion 3.1, then we develop the details of the method in Section 3.2.

### 3.1. Problem setting

**Definition 1 (Language generative model).** A Language generative model $M : X \rightarrow Y$ maps any input prompt $x \in X$ to an output $y \in Y$, where $X$ the prompt space, $Y$ the output space. We denote $Y_h$ is human written text space, $Y_u$ is the machine generated unwatermarked text, $Y_w$ is the machine generated watermarked text.

**Definition 2 (Watermark Algorithm):** A watermark algorithm consists of a watermarking function $W$, a secret key $k$, and a detector $D$. The watermarking function $W$, parameterized by the key $k$, denoted as $W_k$, modifies the output $y$ to embed a watermark, given an input prompt $x \in X$ resulting in a watermarked output $y_w$ which $M(x, W_k) \rightarrow y_w \in Y_w$. The detector $D$, using the same key $k$, can then verify whether a given output $\hat{y} \in Y$ contains the embedded watermark. The detector $D$ operates as a binary classifier with the following output behavior:

$$D(W_k, \hat{y}) = \begin{cases} 1 & \text{if } \hat{y} \text{ is detected as watermarked} \\ 0 & \text{otherwise} \end{cases} \quad (1)$$

The detector $D$ contains a parameter $\theta$, where the $\theta$ is the z-score threshold.

**Definition 3 (Perturbation Function):** The attacker has a perturbation function $P : Y_w \rightarrow Y_p$ modifies the watermarked output $y_w$ to produce a perturbed output $y_p = P(y_w)$. The function $P$ aims to minimize the detection success rate of the detector $D$ on the perturbed output $y_p$. A function $S(y_w, y_p)$ measures the semantic similarity between the original watermarked output $y_w$ and the perturbed

output $y_p = P(y_w)$. The pre-set threshold $\epsilon \in [0, 1]$ is a parameter that quantifies the minimum required level of semantic similarity between the original watermarked output $y_w$ and the perturbed output $y_p = P(y_w)$.

**Assumption**: We define the scenario as a **black box adversarial problem** and we assume that the attacker **should not know the watermark algorithm $W$, the secret key $k$ and should not have access to the detector $D$**. The attacker does not have access to any information about the feature distribution of the watermark algorithm or the model architecture.

For watermark algorithm, the goal is to achieve a balance between robustness and performance. The detector $D$ is formulated as an optimization problem with the objective of minimizing classification errors. Specifically, the detector aims to maximize its accuracy in distinguishing between human-written text $y_h$ and watermarked text $y_w$. The goal of detector $D$ can represente as:

$$\max_{\theta_D} \quad \mathbb{E}_{y_h \sim Y_h} \left[ \log \left( 1 - D_{\theta_D}(W_k, y_h) \right) \right]$$
$$+ \mathbb{E}_{y_w \sim Y_w} \left[ \log \left( D_{\theta_D}(W_k, y_w) \right) \right] \quad (2)$$

For attacker, the perturbation function $P$ is defined to minimize the probability that the detector $D$ successfully identifies the watermark in the perturbed output $y_p$, while ensuring semantic preservation. The goal for $P$ can represente as:

$$P^* = \arg\min_P \quad \mathbb{E}\left[ D(W_k, P(y_w)) \right] \quad (3)$$
$$\text{s.t.} \quad S(y_w, P(y_w)) \geq \epsilon \quad (4)$$

Note that $D(W_k, P(y_w))$ is only used during the evaluation phase. The attacker does not have access to the detector during the training or generation stages.

## 3.2. Self-information rewrite attack

A primary challenge in watermark removal attacks is identifying the "green token" defined by the watermarking algorithm. Some methods, such as Random Walk (Zhang et al., 2023), use grammatical group matching to explicitly replace verbs. In contrast, approaches like DIPPER (Krishna et al., 2024) and GPT Paraphraser (Liu et al., 2024) delegate the task of rewriting and removing green token to large language models through high-level instructions. However, methods of this type lack transparency and control; relying on LLM for consistency with original watermarked text.

Our attack is based on a common principle of watermarking algorithms, as discussed in the KGW (Kirchenbauer et al., 2023; Liu et al., 2023) work: since the watermark must remain imperceptible to the user, high-entropy tokens are ideal candidates for embedding. High-entropy tokens exhibit a more uniform distribution of probabilities, this uniformity means that when logits are adjusted to increase the likelihood of green tokens, it is easier to embed watermarks effectively without significantly compromising the quality of the output. Meanwhile this also implied high-entropy token has lower probability thus higher self-information.

In our approach, we propose a straightforward and easily implementable solution by leveraging self-information to identify potential green-list tokens and subsequently rewrite them. High-entropy tokens are typically associated with high self-information due to their unpredictability and low probability of occurrence. Meanwhile, small probability changes caused by the watermark algorithm can reduce self-information. By considering both the change in self-information and high-entropy token inherent nature, we classify tokens with high or moderate self-information as potential green-list tokens and filter them out to obtain a more neutral template for LLM rewriting. Empirically, our preliminary experiments show that utilizing self-information, rather than directly filtering based on high entropy, results in higher attack success rates. We present a detailed discussion in Appendix G.

Given a watermarked text $y = \{y_0, y_1, \ldots, y_n\}$, where $y_i$ represents each token, we employ a base language model $M_{attack}$; $M_{attack}$ is distinct from the generative model $M$ used to produce the watermarked text. We use $M_{attack}$ to calculate the self-information for each token $y_t$ as follows:

$$I(y_t) = -\log P(y_t | y_0, y_1, \ldots, y_{t-1}; M_{attack}),$$

where $P(y_t | y_0, y_1, \ldots, y_{t-1}; M_{attack})$ denotes the probability of token $y_t$ given its preceding tokens in the sequence, as estimated by the language model $M_{attack}$. To mask the potential green list tokens, we set a threshold $\epsilon$, and get the overall paragraph threshold by percentile:

---

**Algorithm 1** Pseudocode for Self-information rewrite attack

1: **Input:** Watermarked token sequence $\mathbf{y} = \{y_1, y_2, \ldots, y_n\}$, language model $M_{attack}$, self-information percentile $\epsilon$, instruction $\mathbf{s}$
2: **Output:** Response token sequence $\mathbf{y_p}$ without watermark.
3: $\mathbf{y}' \leftarrow M_{attack}(\mathbf{y})$ ▷ Paraphrase sequence $\mathbf{y}'$ using $M_{attack}$
4: $\mathbf{I} \leftarrow [\,]$
5: **for** $i = 1$ to $n$ **do** ▷ Compute self-information for each token in $\mathbf{y}$
6:     $\mathbf{I}[i] \leftarrow -\log P(y_i \mid \text{context})$
7: **end for**
8: $\tau_\epsilon \leftarrow \text{Percentile}(\mathbf{I}, \epsilon)$ ▷ Determine threshold from $\epsilon$ percentile of $\mathbf{I}$
9: **for** $i = 1$ to $n$ **do**
10:     **if** $\mathbf{I}[i] > \tau_\epsilon$ **then**
11:         $y_i \leftarrow \varnothing$ ▷ Mask token if above threshold
12:     **end if**
13: **end for**
14: $\mathbf{y}_p \leftarrow M_{attack}(\mathbf{y}', \mathbf{y}, \mathbf{s})$ ▷ Generate de-watermarked response $\mathbf{y_p}$ using $M_{attack}$
15: **return** $\mathbf{y_p}$

---

$$\tau_\epsilon \leftarrow \text{Percentile}(\mathbf{I}, \epsilon)$$

Any token with a self-information value $\mathbf{I}[i] > \tau_\epsilon$ is considered to be a potential token and will be masked and replaced with a placeholder. In our experiments, we discovered that using placeholders outperformed directly masking specific tokens. The placeholders serve as cues, maintaining the text's structure, indicating where tokens have been masked which providing the LLM with hints about the original text's length and the likely number of words, allowing for more high quality reconstructions.

However, the compression will still result in the loss of watermark text information details. To address this, we use the base LLM $M_{attack}$ to rewrite the watermarked text, creating a reference text. This rewritten text serves as a reference to preserve semantic integrity during the second step. The reason we do not use the original watermarked text is that we find this leads LLM to take shortcuts: LLM tend to directly take the content from the watermark text, due to the high similarity between masked and watermark text.

In the final attack phase, we provide the $M_{attack}$ with the masked text, reference text, and instructions for a fill-in-the-blank task, guiding it to reconstruct the missing content with greedy decoding strategy. We provide the instructions we use in Appendix E. The pseduocode of our algorithm is shown in Algorithm 1.

*Table 1.* Comparison of watermark algorithms under different attack methods. The best results are marked in **bold** and the second best results are marked in underline. Our most lightweight method outperforms all previous paraphrasing attacks. SIRA-Large achieves 100% or near 100% attack success rates on all seven tested watermarking algorithms under black-box settings.*Due to differing threat models, we can not conduct a fair comparison with informed watermark attack methods, thus these methods are excluded* .

| Comparison of Watermark Algorithms under Different Attack Methods | | | | | | | |
|---|---|---|---|---|---|---|---|
| **Watermark** 
 **Attack** | **KGW-1** | **Unigram** | **UPV** | **EWD** | **DIP** | **SIR** | **EXP** |
| **Word delete (Welbl et al., 2020)** | 22.4% | 1.6% | 6.6% | 22.8% | 57.4% | 44.0% | 9.4% |
| **Synonym Substitution (Yu et al., 2010)** | 83.2% | 17.4% | 65.2% | 76.2% | 99.6% | 82.0% | 51.0% |
| **GPT Paraphraser** | **100%** | 63.9% | 71.9% | 90.8% | **99.8%** | 58.8% | 72.2% |
| **DIPPER-1 (Krishna et al., 2024)** | 82.4% | 37.0% | 58.6% | 82.2% | 99.6% | 61.2% | 73.6% |
| **DIPPER-2 (Krishna et al., 2024)** | 95.8% | 45.6% | 61.8% | 89.0% | **99.8%** | 63.6% | 82.2% |
| **SIRA-Tiny(Ours)** | 96.4% | 87.6% | 84.4% | 97.8% | 99.8% | 75.0% | 90.6% |
| **SIRA-Small(Ours)** | **100%** | 93.8% | 93.0% | **100%** | 99.8% | 83.4% | 93.4% |
| **SIRA-Large(Ours)** | **100%** | **100%** | **99.6%** | **100%** | **100%** | **98.8%** | **99.8%** |

## 4. Experiments

### 4.1. Setup

**Dataset and Prompts.** Following prior watermarking research (Kirchenbauer et al., 2023; Zhao et al., 2023; Liu et al., 2024; Kuditipudi et al., 2023), we utilize the C4 dataset (Raffel et al., 2020a) for general-purpose text generation scenarios. We selected 500 random samples from the test set to serve as prompts for generating the subsequent 230 tokens, using the original C4 texts as non-watermarked examples.

**Watermark generation algorithms and language model.** To conduct a comprehensive evaluation, we select seven recent watermarking works: KGW (Kirchenbauer et al., 2023), Unigram (Zhao et al., 2023), UPV (Liu et al., 2023), EWD (Lu et al., 2024), DIP (Wu et al., 2024), SIR (Liu et al., 2024), EXP (Aaronson & Kirchner, 2022) in the assessment. The watermark hyperparameter settings shown in Appendix A, and the detection settings adhere to the default/recommendations (Pan et al., 2024) configurations of the original works. Specifically, for KGW-k, $k$ is the number of preceding tokens to hash. A smaller $k$ implies stronger attack robustness yet simpler watermarking rules. We use KGW-1 in our experiment. For language models, we follow the previous work setting select (Kirchenbauer et al., 2023; Liu et al., 2024; Zhao et al., 2023) Opt-1.3B (Zhang et al., 2022) as the watermark text generation model. Our SIRA Tiny,SIRA Small, and SIRA Large run on the Llama3 Instruct models (Dubey et al., 2023) with 3B, 8B, and 70B parameters, respectively.

**Baseline Methods.** For our method, we use $\epsilon = 0.3$ as threshold. For the attack method, we use word deleteion (Welbl et al., 2020), synonym substitution (Yu et al., 2010), Dipper (Krishna et al., 2024), and GPT Paraphaser (Liu et al.,

2024) to compare with our method. For GPT Paraphaser, we use the GPT-4o-2024-05-13 (OpenAI, 2024) version. For DIPPER-1 the lex diversity is 60 without order diversity, and for DIPPER-2 we additionally increase the order diversity by 40. The word deletion ratio is set to 0.3 and the synonym substitution ratio is set to 0.5. The synonyms are obtained from the WordNet synset (Miller, 1995).

**Evaluation.** We utilize the attack success rate as our primary metric. The attack success rate is defined as the proportion of generated attack texts for which the watermark detector incorrectly classify the attack text as the unwatermarked sample, compared to the total number of attack texts. To mitigate the influence of detection thresholds, we follow prior work (Liu et al., 2024; Zhao et al., 2023) adjust z-threshold of detector until reaches target false positive rate in Figure 2 . We used generated 500 attack texts as positive samples and 500 human-written texts as negative samples. We dynamically adjust the detector's thresholds to establish false positive rates at 1% and 10%, and we report the true positive rates and F1-scores. Our method runs on NVIDIA A100 GPUs(Tiny, Small run on a single GPU).

### 4.2. Experimental Results.

In Table 1, we present the attack success rates of various watermark removal methods across different watermarking algorithms. The results demonstrate that our approach consistently outperforms all other methods for each watermarking algorithm. Notably, the closest competitors to our method are DIPPER (Krishna et al., 2024) and GPT Paraphraser, which are model-based paraphrasing attacks. **Even Our most lightweight SIRA-Tiny method outperforms all previous approaches** regarding attack success rate in our experiments involving seven watermarking algorithms on the C4 dataset (Raffel et al., 2020a). Our Large

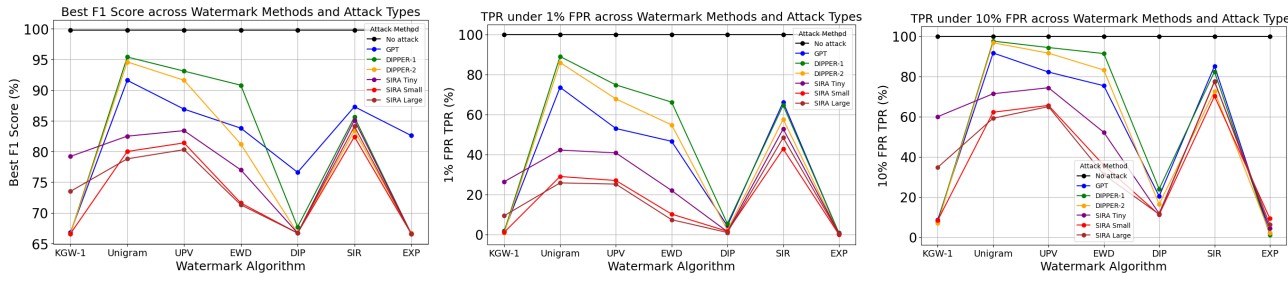

(a) True positive rate with FPR set to 1%.   (b) True positive rate with FPR set to 10%.   (c) Best F1-score achieved by watermark.

*Figure 2.* To mitigate the default z-threshold's impact on the robustness of watermarking algorithms, we dynamically adjust the z-score threshold until the watermark detector achieves specified false positive rates. The true positive rate (TPR↓) and the best F1 score are shown. Lower TPR and F1 scores at a given false positive rate (FPR) indicate that the watermark detector struggles to distinguish attack texts from human-written texts, suggesting a more effective attack. Detailed values for the figures are provided in Appendix C.

method achieves **nearly 100% attack success** across all seven tested watermarking algorithms.

To further demonstrate the effectiveness of our method and avoid the impact of a fixed z-threshold on detector performance, we follow previous work by setting the FPR to 1% and 10%, and report the true positive rate of the detector on adversarial texts based on the adjusted z-threshold corresponding to the FPR. Additionally, we report the best F1 score that the watermark algorithm can achieve under different attacks. The results are shown in Figure 2, and the detailed numbers are provided in Appendix C. Lower true positive at a given false positive rate indicate that the watermark detector struggles more to differentiate between adversarial texts and human-written texts. Our algorithm achieves optimal attack performance in most cases; suggesting a more effective attack.

### 4.3. Text quality analysis

To further demonstrate that our method does not adversely affect text quality, we conduct additional evaluations of the text generated by the model. We compare **Perplexity (PPL)** of the text quality. Furthermore, we use a well-established metric sentence-level embedding similarity (**Sentence-BERT Score (s-BERT)**) before and after the attack to explore whether the attack alters the semantic content. We also conducted experiments in the Appendix F using ChatGPT as a judge to measure overall semantic similarity. The results, shown in Figure 3, indicate that our method has a smaller impact on text quality compared to other approaches.Our approach, similar to other model-based methods, benefits from more powerful large language models, achieving better performance in terms of the PPL metric compared to the original watermarked text. Additionally, our method retains a greater degree of semantic information. We show the detail numbers of two metrics in Appendix D.

### 4.4. Ablation Experiment

In this section, we aim to further scrutinize the self-information rewrite attack and emphasize the potential of this attack. We utilize Opt-1.3b and a random sample of 50 prompts from the C4 dataset to generate watermarked responses. Unless otherwise specified, we use Llama-3-8b (SIRA-Small) as the base model for our attack. The temperature for the base model is set to 0.7.

**How does the self-information threshold affect final performance?** In this experiment, we use UPV as the watermarking algorithm. We varied the value of $\epsilon$ from 0.25 to 0.70 in increments of 0.05 to test its impact on the success rate of the attack using the UPV algorithm. The results are shown in Table 2.

We observed that the attack success rate and sentence-bert is directly influenced by the value of $\epsilon$. For the UPV algorithm, setting the threshold to 0.3 results in a highly effective attack. A significant performance gap is observed when $\epsilon$ increases from 0.60 to 0.65. Additionally, there is a tradeoff between attack effectiveness and semantic preservation. When $\epsilon$ is below 0.25, the generated attack text tends to lose more detailed information from the original watermarked text. Considering both performance and semantic preservation, we recommend setting $\epsilon$ between 0.2 and 0.3. For less robust algorithms, setting $\epsilon$ between 0.4 and 0.5 is sufficient to achieve an attack success rate exceeding 90%. Setting $\epsilon$ to 0.3 effectively removes the watermark while preserving the original semantics.

**Self-information mask versus Random mask and Iterative Paraphrase(twice)** In this experiment, we replace self-information-based selective masking with a random masking strategy and Iterative Paraphrase(twice), while keeping all other steps unchanged. We use the same masking ratios, ranging from 0.4 to 0.8 in increments of 0.1, and compare the resulting attack success rates. The Unigram watermark-

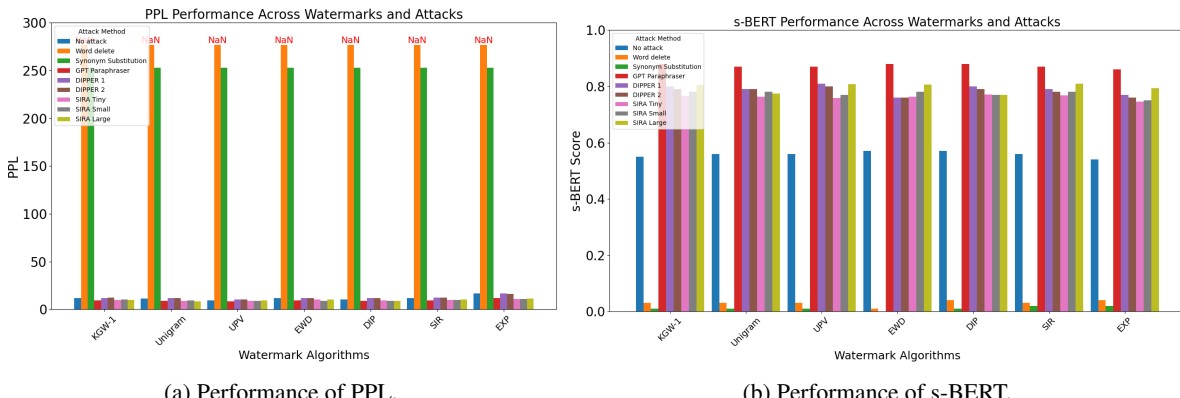

(a) Performance of PPL.

(b) Performance of s-BERT.

*Figure 3.* Performance comparison of watermark methods against various attack methods based on PPL (Perplexity↓) and s-BERT (Sentence-BERT score↑). The word delete will significantly increase the PPL and lead to overflow. We marked the overflow data with NaN in the Figure 3a. The synonym substitution will also increase the PPL. The paraphrased text has better text quality than the original watermark text for our method and GPT Paraphraser. The detailed number are shown in Appendix D.

*Table 2.* Effect of self-information Threshold on the Attack Success Rate and Semantic Preservation of the UPV Algorithm

| self-information threshold $\epsilon$ | 0.25 | 0.30 | 0.35 | 0.40 | 0.45 | 0.5 | 0.55 | 0.60 | 0.65 | 0.70 |
|---|---|---|---|---|---|---|---|---|---|---|
| Attack Success Rate | 96% | 94% | 94% | 88% | 80% | 76% | 72% | 70% | 58% | 32% |
| Sentence-BERT | 0.68 | 0.77 | 0.78 | 0.78 | 0.79 | 0.79 | 0.82 | 0.81 | 0.83 | 0.86 |

*Table 3.* Comparison with Random Masking Strategy and Iterative Paraphrase. Iterative Paraphrase is independent of the mask ratio. For clearer comparison, we place its results under the 0.7 (default mask ratio) column. Notice here the random mask performance also benefits from other steps like rewriting in our framework. The vanilla random mask has a similar attack success rate as word deletion.

| Mask Ratio | 0.4 | 0.5 | 0.6 | 0.7 | 0.8 |
|---|---|---|---|---|---|
| Iterative Paraphrase | - | - | - | 56% | - |
| Random Mask | 52% | 66% | 78% | 80% | 82% |
| Self-information Mask | 80% | 88% | 92% | 96% | 100% |

*Table 4.* Comparison of Attack Success Rate and Average z-score. The reference text is generated by asking the base model to paraphrase the watermarked response, while the attack text is generated using our two-step approach.

| Text | Attack Success Rate | Average z-score |
|---|---|---|
| Human-written Text | N/A | 0.12 |
| Reference Text | 64% | 3.75 |
| Attack Text | 94% | 1.85 |

ing method is employed to generate the watermarked text. The results are presented in Table 3. To ensure fair comparisons, the random masking strategy is executed five times, and the final average attack success rate is reported.

The results indicate that, at any given mask ratio, the self-information-based masking method significantly outperforms the random strategy. The random masking approach also has a bottleneck, with limited improvement in attack success rates beyond a ratio of 0.6. This is due to the random mask not make sure all target green tokens are removed. Also,This experiment ruled out the possibility that the attack effectiveness is primarily caused by the double paraphrasing process. For a single watermarked text with fixed mask ratio, our method is deterministic, as the same tokens are masked each time. In contrast, the random approach does not provide this guarantee.

**Does the success of the attack due to paraphrased reference text?** We used the Unigram watermarking algorithm to

generate watermarked text. We set the detector's z threshold to 4 according to its default settings. For a given input, the detector calculates its z-score, and if the score exceeds 4, the text is classified as watermarked.

We measured the attack success rate for each of the following stages: the reference text generated in the first step of our algorithm, and the final attack text. Additionally, we calculated the average z-score for each stage and reported the z-score of human-written text as a reference. The result are shown in Table 4. We observed that the attack success rate for the reference text is lower than that of the final attack text. Paraphrase strategies tend to preserve more n-grams from the original text, which may still be detectable by the watermark detection algorithm. In contrast, our attack reduces the presence of such n-grams by utilizing self-information filtering. Additionally, the z-score produced by our method is closer to that of human-written text compared to simple paraphrasing approaches.

## 5. Conclusion

In this paper, we present the Self-Information Rewrite Attack (SIRA), a lightweight and effective method for removing watermarks from LLM-generated text by targeting anomalous tokens. Empirical results show that SIRA outperforms existing methods in attack success rates across multiple watermarking techniques while preserving text quality and requiring minimal computational resources. By exploiting vulnerabilities in current watermarking algorithms, SIRA highlights the need for more robust and adaptive watermarking approaches in watermark embedding. We will release our code to the community to facilitate further research in developing responsible AI practices and advancing the robustness of watermarking algorithms.

## Impact Statement

In this work, we aim to provide an approach to test the robustness of Large Language Models watermark. We propose a method that can remove different watermarks in LLM-generated text. We are aware of the potential risks that our work entails for the security and safety of LLMs, as they are increasingly adopted in various domains and applications. Nevertheless, we also believe that our work advances open and transparent research on the challenges and limitations of the LLM watermark, which is crucial for devising effective solutions and protections. Similarly, the last few years the exploration of adversarial attacks (Wei et al., 2023; Madry et al., 2017; Krishna et al., 2024) has led to the improvement of responible AI and led to techniques to safeguard against such vulnerabilities,e further coordinated with them before publicly releasing our results. We also emphasize that, our ultimate goal in this paper is to identify the weaknesses of existing methods.

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

# Contents of the appendix

The contents of the supplementary material are organized as follows:

- In Appendix A, we list the hyperparameters of the watermarking algorithm we used in experiment Section 4.

- In Appendix B, we perform the comparison of execution time and VRAM consumption between our algorithm and other baseline methods.

- In Appendix C, we present the specific data points corresponding to the figure shown in Section 4.2.

- In Appendix D, we provide the precise data underlying the figure depicted in Section 4.3.

- In Appendix E, we provide the prompt we used to generate attack text.

- In Appendix F, we conducted extensive experiments to evaluate the overall preservation of the semantic meaning of the original watermarked text.

- In Appendix G, we offer a brief discussion about the change in self-information under the influence of the watermark algorithm.

- In Appendix H, we provide the full proof of our proposed method attack success rate upper bound and lower bound, together with the proof of lemma.

- In Appendix I, we provide a visual comparison of the text generated by our method with watermarked text, non-watermarked text, and text generated by other attack methods.

- In Appendix J, we provide extended experiments, including a comparison between our methods and baseline approaches on the OpenGen dataset, as well as the performance of our methods against Adaptive Watermark and Waterfall Watermark schemes.

# A. Watermark algorithm setting

In this section, we list the hyperparameters of the watermarking algorithm we used in Section 4 below.

```
1  {
2      "algorithm_name": "KGW",
3      "gamma": 0.5,
4      "delta": 2.0,
5      "hash_key": 15485863,
6      "prefix_length": 1,
7      "z_threshold": 4.0
8  }
```

*Listing 1.* configuration KGW

```
1  {
2      "algorithm_name": "Unigram",
3      "gamma": 0.5,
4      "delta": 2.0,
5      "hash_key": 15485863,
6      "z_threshold": 4.0
7  }
```

*Listing 2.* configuration Unigram

```
1  {
2      "algorithm_name": "UPV",
3      "gamma": 0.5,
4      "delta": 2.0,
5      "z_threshold": 4.0,
6      "prefix_length": 1,
7      "bit_number": 16,
8      "sigma": 0.01,
9      "default_top_k": 20,
10 }
```

*Listing 3.* configuration UPV

```
1  {
2      "algorithm_name": "EWD",
3      "gamma": 0.5,
4      "delta": 2.0,
5      "hash_key": 15485863,
6      "prefix_length": 1,
7      "z_threshold": 4.0
8  }
```

*Listing 4.* configuration EWD

```
1  {
2      "algorithm_name": "DIP",
3      "gamma": 0.5,
4      "alpha":0.45,
5      "hash_key": 42,
6      "prefix_length": 5,
7      "z_threshold": 1.513,
8      "ignore_history": 1
9  }
```

*Listing 5.* configuration DIP

```
1  {
2      "algorithm_name": "SIR",
3      "delta": 1.0,
4      "chunk_length": 10,
5      "scale_dimension": 300,
6      "z_threshold": 0.0,
7  }
```

*Listing 6.* configuration SIR

```
1  {
2      "algorithm_name": "EXP",
3      "prefix_length": 4,
4      "hash_key": 15485863,
5      "threshold": 2.0,
6      "sequence_length": 230
7  }
```

*Listing 7.* configuration EXP

## B. Execution time and VRAM consumption comparison

In this section, We conducted the attack experiments using 50 distinct watermark texts, each containing approximately 230 ± 20 tokens. For each method, we measured both execution time and VRAM usage. The reported execution time reflects the average for a single attack instance. The experiments were run on NVIDIA A100 40GB GPUs, utilizing a sequential device map for baseline methods requiring multiple GPUs. The configuration for the GPT Paraphraser follows the setup described in Section 4.1. The results are shown in Table 5.

One of the main limitations of current model-based watermark removal attacks is their substantial resource consumption. For instance, DIPPER built on the T5-XXL model, necessitates two 40GB A100 GPUs for effective operation. Similarly, the GPT parser introduces considerable costs due to its dependence on a proprietary model that employs token-based billing.The total time consumption for SIRA consists of two parts: two generations by the base model and the self-information mask. Using SIRA-Tiny as example, The self-information mask is nearly negligible, as it does not require any text generation (less than 0.1 seconds). The other two generations take around 2.6 seconds per generation on a single A100 GPU. Thus the total execution time is around 10 seconds. We use the huggingface library in our experiment.

The DIPPER method utilizes a specially fine-tuned T5-XXL model for text paraphrasing. This model needs at least 40 GB VRAM to run and one-time generation requires around 15 seconds on two A100GPU.

Our proposed pipeline operates with a minimal configuration of the LLaMA3-3b model, which is lightweight enough to run on mobile-level devices. This ensures compatibility with many consumer-grade GPUs, significantly reducing hardware requirements. Notably, even our most lightweight approach, SIRA-Tiny, outperforms all previous methods in our experiments while consuming far fewer resources.

Moreover, our method is independent of specific LLM architectures, allowing it to be seamlessly transferred to the latest language models to leverage their superior performance at no additional cost. As demonstrated in Table 5, the data was generated using bf16; our VRAM consumption can be further reduced by employing mixed precision or quantized models, further enhancing efficiency.

*Table 5.* Comparison of Execution Time and VRAM Usage for Different Methods.

| Method | Execution Time (s) | VRAM Usage (GB) |
|---|---|---|
| GPT Paraphraser | 12.8 | N/A |
| DIPPER | 14.7 | 44.56 |
| SIRA-Tiny | 5.3 | 7.06 |
| SIRA-Small | 10.3 | 18.20 |
| SIRA-Large | 37.2 | 138.60 |

We also provide a comprehensive cost analysis of our method in comparison to third-party paraphrasing services. Specifically, we estimate the cost of processing 1 million tokens of watermarked text. According to OpenAI's pricing, paraphrasing using GPT-4o (GPT Paraphraser) costs

$$\$0.01 \times 2 \times 10 = \$20,$$

where $0.01 is the cost per 1M tokens (input or output), the factor 2 accounts for both input and output, and 10 iterations are assumed.

In contrast, our method (SIRA-Small), based on LLaMA3-8B deployed via AWS Bedrock, incurs a significantly lower cost:

$$\$0.22 \times 2 \times 2 = \$0.88,$$

where $0.22 is the per-1M-token cost (input or output), the first factor of 2 accounts for input and output, and the second factor of 2 accounts for two rewriting iterations. The cost can be further reduced using SIRA-Tiny (LLaMA3-3B).

## C. Best F1 score and TPR/FPR

In Table 6, we list the specific data from the figure in Figure 2, which reflects different attack method's performance of dynamically adjusting the watermark detector's z-threshold until a specified false positive rate is achieved. We report both the F1 score and the true positive rate. It can be observed that, in most cases, our attack method achieves the best performance.

*Table 6.* In this experiment, we dynamically adjust the z-score threshold of the watermarking algorithm until achieving specified false positive rates for the watermark detector. Lower TPR and F1 scores indicate that the watermark detector struggles more to differentiate between attack texts and human-written texts, suggesting a more effective attack.

| Method | Attack Type | 1% FPR | | 10% FPR | | Best F1 (%) |
|---|---|---|---|---|---|---|
| | | TPR (%) | F1 (%) | TPR (%) | F1 (%) | F1 (%) |
| | No attack | 100 | 99.5 | 100 | 95.2 | 99.8 |
| | DIPPER -1 | 1.6 | 3.1 | 7.8 | 13.3 | **66.6** |
| | DIPPER -2 | **0.8** | **1.6** | **7.0** | **12.1** | **66.6** |
| KGW-1 | GPT Paraphraser | 1.8 | 3.6 | 8.7 | 14.0 | 66.8 |
| | SIRA Tiny | 26.4 | 41.4 | 60.0 | 70.6 | **79.2** |
| | SIRA-Small | 1.1 | 2.1 | 8.4 | 14.0 | **66.6** |
| | SIRA Large | 1.1 | 2.1 | 8.0 | 13.6 | **66.6** |
| | No attack | 100 | 99.5 | 100 | 95.2 | 99.8 |
| | DIPPER -1 | 89.0 | 93.7 | 97.6 | 95.8 | 95.4 |
| | DIPPER -2 | 86.0 | 91.8 | 96.8 | 94.2 | 94.6 |
| Unigram | GPT Paraphraser | 73.6 | 84.3 | 91.7 | 91.6 | 91.6 |
| | SIRA Tiny | 42.2 | 58.9 | 71.4 | 78.7 | 82.5 |
| | SIRA-Small | 29.0 | 44.6 | 62.2 | 72.7 | 80.0 |
| | SIRA Large | **25.8** | **40.7** | **59.2** | **69.9** | **78.8** |
| | No attack | 100 | 99.5 | 100 | 95.2 | 99.8 |
| | DIPPER -1 | 74.8 | 85.2 | 94.4 | 93.1 | 93.1 |
| | DIPPER -2 | 67.8 | 80.4 | 91.6 | 91.6 | 91.6 |
| UPV | GPT Paraphraser | 53.0 | 69.2 | 82.2 | 86.2 | 86.9 |
| | SIRA Tiny | 40.8 | 57.5 | 74.4 | 80.7 | 83.4 |
| | SIRA-Small | 27.0 | 42.3 | 65.6 | 75.4 | 81.4 |
| | SIRA Large | **25.2** | **39.9** | **65.0** | **74.3** | **80.3** |
| | No attack | 100 | 99.5 | 100 | 95.2 | 99.8 |
| | DIPPER -1 | 66.2 | 79.2 | 91.4 | 90.8 | 90.8 |
| | DIPPER -2 | 54.8 | 70.3 | 83.2 | 81.2 | 81.2 |
| EWD | GPT Paraphraser | 46.6 | 63.1 | 75.4 | 81.3 | 83.8 |
| | SIRA Tiny | 22.0 | 35.8 | 52.2 | 64.4 | 77.0 |
| | SIRA-Small | 10.2 | 18.3 | 35.8 | 49.1 | 71.6 |
| | SIRA Large | **7.4** | **13.7** | **31.2** | **44.2** | **71.3** |
| | No attack | 100 | 99.5 | 100 | 95.2 | 99.8 |
| | DIPPER -1 | 5.4 | 10.0 | 24.0 | 36.1 | 67.7 |
| | DIPPER-2 | 2.2 | 4.3 | 16.4 | 26.2 | **66.7** |
| DIP | GPT Paraphraser | 4.3 | 8.3 | 20.4 | 32.0 | 76.6 |
| | SIRA Tiny | 1.4 | 2.7 | 11.8 | 19.4 | **66.7** |
| | SIRA-Small | 1.6 | 3.1 | 11.2 | 18.7 | **66.7** |
| | SIRA Large | **1.0** | **1.9** | **11.4** | **18.8** | **66.7** |
| | No attack | 100 | 99.5 | 100 | 95.2 | 99.8 |
| | DIPPER -1 | 64.6 | 78.0 | 82.4 | 85.6 | 85.6 |
| | DIPPER -2 | 57.6 | 72.6 | 72.6 | 83.4 | 83.4 |
| SIR | GPT Paraphraser | 66.2 | 79.2 | 85.2 | 87.3 | 87.3 |
| | SIRA Tiny | 52.8 | 68.7 | 77.6 | 82.7 | 85.1 |
| | SIRA-Small | **42.8** | **59.5** | **70.2** | **77.9** | **82.4** |
| | SIRA Large | 48.4 | 64.8 | 77.4 | 82.6 | 84.1 |
| | No attack | 100 | 99.5 | 100 | 95.2 | 99.8 |
| | DIPPER -1 | 0.8 | 1.6 | 1.2 | 2.1 | **66.6** |
| | DIPPER -2 | 0.4 | 0.8 | **2.0** | **3.8** | 66.7 |
| EXP | GPT Paraphraser | 0.4 | 0.8 | **2.0** | **3.8** | 82.6 |
| | SIRA Tiny | 0.6 | 1.2 | 4.4 | 7.7 | **66.6** |
| | SIRA-Small | **0.4** | **0.8** | 9.3 | 15.6 | **66.6** |
| | SIRA Large | **0.4** | **0.8** | 6.2 | 10.7 | **66.6** |

## D. Detail number of PPL and sentence bert score

In this section, we list the detail number of PPL and sentence-bert score we present in the Section 4.3.

| | KGW-1 | | Unigram | | UPV | | EWD | | DIP | | SIR | | EXP | |
|---|---|---|---|---|---|---|---|---|---|---|---|---|---|---|
| Attack | PPL(↓) | s-BERT(↑) | PPL(↓) | s-BERT(↑) | PPL(↓) | s-BERT(↑) | PPL(↓) | s-BERT(↑) | PPL(↓) | s-BERT(↑) | PPL(↓) | s-BERT(↑) | PPL(↓) | s-BERT(↑) |
| No attack | 12.00 | 0.55 | 11.49 | 0.56 | 9.27 | 0.56 | 11.64 | 0.57 | 10.60 | 0.57 | 11.76 | 0.56 | 16.48 | 0.54 |
| Word delete | NaN | 0.03 | NaN | 0.03 | NaN | 0.03 | NaN | 0.01 | NaN | 0.04 | NaN | 0.03 | NaN | 0.04 |
| Synonym Substitution | 252.85 | 0.01 | 252.85 | 0.01 | 252.85 | 0.01 | 252.85 | 0.00 | 252.85 | 0.01 | 252.85 | 0.02 | 252.85 | 0.02 |
| GPT Paraphraser | 9.19 | 0.88 | 8.96 | 0.87 | 8.28 | 0.87 | 9.20 | 0.88 | 8.79 | 0.88 | 9.52 | 0.87 | 11.98 | 0.86 |
| DIPPER-1 | 12.00 | 0.80 | 11.80 | 0.79 | 10.31 | 0.81 | 11.87 | 0.76 | 11.93 | 0.80 | 12.43 | 0.79 | 16.56 | 0.77 |
| DIPPER-2 | 12.15 | 0.79 | 11.80 | 0.79 | 10.34 | 0.80 | 11.96 | 0.76 | 11.86 | 0.79 | 12.42 | 0.78 | 16.45 | 0.76 |
| SIRA-Tiny (Ours) | 9.97 | 0.77 | 9.14 | 0.76 | 8.93 | 0.76 | 10.45 | 0.76 | 9.43 | 0.77 | 10.09 | 0.77 | 10.96 | 0.75 |
| SIRA-Small (Ours) | 10.59 | 0.78 | 9.37 | 0.78 | 8.91 | 0.77 | 9.10 | 0.78 | 9.02 | 0.77 | 9.74 | 0.78 | 10.93 | 0.75 |
| SIRA-Large (Ours) | 10.14 | 0.80 | 8.62 | 0.77 | 9.39 | 0.81 | 10.22 | 0.81 | 8.93 | 0.77 | 10.35 | 0.81 | 11.32 | 0.79 |

*Table 7.* Performance of different attack methods against various watermark techniques, including SIRA variants.

## E. Fill in blank instruction

We present our generation prompts(final step) and reference text generation prompts below. The reference text generation followed the same prompt of (Sadasivan et al., 2023). During the generation process, the reference text and the masked text are replaced by the two texts obtained in the first step.

> **Generation Prompts**
>
> You will be shown one reference paragraph and one incomplete paragraph. Your task is to write a complete paragraph using incomplete paragraph. The complete paragraph should have similar length with reference paragraph. You need to include all the information in the reference. but do not take the expression and words in the reference paragraph. You should only answer the complete paragraph.
> reference: [reference text]
> incomplete paragraph: [mask text]

> **Reference Parapharse Prompts**
>
> You are a paraphraser. You are given an input passage 'INPUT'. You should paraphrase 'INPUT' to print 'OUTPUT'. 'OUTPUT' should be diverse and different as much as possible from 'INPUT' and should not copy any part verbatim from 'INPUT'. However, 'OUTPUT' should preserve the information in the INPUT. You should print 'OUTPUT' and nothing else so that it is easy for me to parse.:

## F. GPT Semantic Judge

> **GPT Semantic Judge Prompts**
>
> You are a helpful assistant. You will receive a original text and a pharased text .Please act as an impartial judge and evaluate how much the pharased text keep the original text semantic meaning. Be as objective as possible. Rate the response on a scale from 1 to 10. A rating of 1 signifies the pharased text has totally different meaning with the original text. A rating of 10 signifies the pharased text has same meaning with the original text . Your answer should strictly be the format: [Rating].
> Here's the original text: [Watermarked text]
> Here's the pharsed text:[Attack text]

In this section, we conduct extensive experiments to evaluate the overall preservation of the semantic meaning of the original watermarked text. We use ChatGPT (OpenAI, 2024) as an impartial judge to obtain the quantitative results.

The attack success rate alone is not a sufficient metric for evaluating an attack method. It is also crucial to assess whether the original and paraphrased outputs preserve similar semantics. The Sentence-BERT score (Reimers & Gurevych, 2019), presented in Section 4.3 , measures the sentence-level similarity between the original watermarked text and the adversarial

text. However, it falls short in determining whether the overall semantics are preserved. Inspired by the LLM jailbreak work PAIR (Chao et al., 2023), which leverages carefully crafted prompts and the powerful capabilities of ChatGPT to score attack texts and targets for quantitative evaluation, we adapted their prompts to use ChatGPT for assessing the semantic similarity between watermarked texts and attack texts . This approach allows us to obtain semantic similarity scores that more closely align with human perception. We show the judge prompt in Appendix F and the result in shown in Table 8.

*Table 8.* Semantic Preservation for Different Methods

| | Word Delete | Synonym | GPT Paraphraser | DIPPER-1 | DIPPER-2 | SIRA-Tiny | SIRA-Small | SIRA-Large |
|---|---|---|---|---|---|---|---|---|
| **Semantic Preservation** | 2.59 | 2.63 | 8.25 | 5.28 | 6.34 | 6.10 | 6.84 | 8.02 |

We observed that using GPT for paraphrasing alone best preserves the original text's semantics, whereas methods like word deletion and synonym replacement were largely ineffective. Our approach demonstrated superior semantic preservation compared to the DIPPER method.

# G. Self-information, Entropy and Probability

We provide a brief explanation of how the watermark algorithm changes the self-information. To begin, we introduce the definitions of self-information.

**Self-Information** ($I(x)$): This measures the amount of information or "surprise" associated with a specific token $x$. It quantifies how unexpected the occurrence of a token is in a given context:

$$I(x) = -\log_2 P(x)$$

When considering the context $h$, it becomes the conditional self-information:

$$I(x \mid h) = -\log_2 P(x \mid h)$$

where $P(x \mid h)$ is the probability of token $x$ occurring given the preceding context $h$.

We first analyze the non-conditional scenario, assuming that watermarking slightly increases the probability of certain tokens by a small amount $\delta$, while adjusting the probabilities of other tokens to maintain normalization. The $\delta$ change in token influenced by watermark algorithm is usually very small (e.g less than 1e-3).

The adjusted probability for the watermarked token $x_w$ is:

$$P'(x_w) = P(x_w) + \delta$$

The adjusted probabilities for other tokens $x_i$ ($i \neq w$) are:

$$P'(x_i) = P(x_i) - \epsilon_i$$

where $\sum_{i \neq w} \epsilon_i = \delta$.

The change in entropy due to these adjustments is given by:

$$\Delta H = H(P') - H(P) = -\sum_i [P'(x_i) \log P'(x_i) - P(x_i) \log P(x_i)]$$

The partial derivative of entropy with respect to $P(x_w)$ is:

$$\frac{\partial H}{\partial P(x_w)} = -\log P(x_w) - 1$$

The change in entropy due to a small change $\delta$ in $P(x_w)$ is approximately:

$$\Delta H \approx \frac{\partial H}{\partial P(x_w)} \delta = -(\log P(x_w) + 1)\delta$$

In high-entropy contexts, where $P(x_w)$ is small, $\log P(x_w)$ becomes a large negative value. Therefore, $\log P(x_w) + 1$ is still negative, and the product with the small $\delta$ results in a tiny $\Delta H$(decrease in logarithmically). **This attribute makes the watermark algorithm need to embed patterns in high-entropy tokens**, otherwise it will significantly compromise the quality of the output.

For self-information, the change in self-information is:

$$\Delta I(x_w) = -\log P'(x_w) + \log P(x_w)$$

The derivative of self-information with respect to $P(x_w)$ is:

$$\frac{dI(x_w)}{dP(x_w)} = -\frac{1}{P(x_w)}$$

For small $P(x_w)$, $\frac{1}{P(x_w)}$ is large, making $\Delta I(x_w)$ more significant for small $\delta$ compared to $\Delta H$.

Similarly, for conditional self-information, The uniform distribution serves as a theoretical upper bound for high entropy for a given probability space and helps illustrate high-entropy scenarios where probability mass is thinly spread. In such cases, we assume that the model predicts $N$ possible next tokens with equal probability. Where:

$$P(x \mid \text{Context}) = \frac{1}{N}$$

For large $N$, $P(x \mid \text{Context})$ becomes small.

The adjusted probability for the watermarked token $x_w$ is:

$$P'(x_w \mid \text{Context}) = \frac{1}{N} + \delta$$

The adjusted probabilities for other tokens $x_i$ $(i \neq w)$ are:

$$P'(x_i \mid \text{Context}) = \frac{1}{N} - \frac{\delta}{N-1}, \quad \text{for } x_i \neq x_w$$

The change in Conditional Self-Information is:

$$\Delta I(x_w \mid \text{Context}) = I'(x_w \mid \text{Context}) - I(x_w \mid \text{Context}) = -\log\left(\frac{1}{N} + \delta\right) + \log N$$

Using a Taylor series approximation for small $\delta$:

$$\log\left(\frac{1}{N} + \delta\right) \approx \log\left(\frac{1}{N}\right) + N\delta$$

The approximate change in conditional self-information is:

$$\Delta I(x_w \mid \text{Context}) \approx -N\delta$$

Compared to the change in entropy, it is obvious self-information are more sensitive metric:

$$\Delta H \approx \frac{\partial H}{\partial P(x_w)}\delta = -(\log P(x_w) + 1)\delta$$

When $P(x \mid \text{Context})$ is small, the magnitude of the derivative is large, this indicates that small changes in $P(x \mid \text{Context})$ result in bigger changes in $I(x \mid \text{Context})$. As a result, the green token influenced by the watermark change will have less self-information than the original.

High-entropy tokens are usually associated with medium,high self-information due to their unpredictability and low probability of occurrence. Considering the reduced self-information, these potential green tokens generally will exhibit high or moderate self-information values. Therefore in practice, we filter out all tokens with high or moderate self-information. This ensures we can comprehensively eliminate potential tokens.

# H. Theoretical Proof

## H.1. Preliminaries and Notation

**Watermarked Text.** Let $y = \{y_1, y_2, \ldots, y_n\}$ be a sequence of $n$ tokens generated by a watermarked language model $M$. A subset $\mathcal{W} \subseteq \{1, 2, \ldots, n\}$ denotes the indices of "green" (watermarked) tokens.

**Base (Attack) Model.** Let $M_{\text{attack}}$ be a language model distinct from $M$. Under $M_{\text{attack}}$, each token $y_i$ has probability

$$P(y_i \mid y_1, \ldots, y_{i-1}; M_{\text{attack}}).$$

**Self-Information.** The self-information of $y_i$ under $M_{\text{attack}}$ is defined as

$$I(y_i) = -\log P\big(y_i \mid y_1, \ldots, y_{i-1}; M_{\text{attack}}\big).$$

A larger $I(y_i)$ indicates $y_i$ is more "surprising" or low-probability under $M_{\text{attack}}$.

**Attack Strategy.**

- **Threshold Selection**: Choose a threshold $\tau$ (e.g., a certain percentile $\tau_\epsilon$) and mask tokens whose self-information exceeds $\tau$.

- **Rewrite Step**: Provide the masked text plus a reference text to an LLM, instructing it to fill the placeholders.

- **Success Criterion**: The attack is considered successful if *all* watermarked tokens are significantly altered so that the watermark is no longer detectable.

## H.2. Lemmas on Self-Information Changes Due to Watermarking

**Lemma H.1** (Bound on Self-Information Shift). *Suppose a watermarking algorithm increases the probability of a single token $x_w$ from $P(x_w)$ to $P'(x_w) = P(x_w) + \delta$, where $\delta$ is small (i.e., $\delta \ll 1$) and $P(x_w) \ll 1$. Let*

$$I(x_w) = -\log P(x_w), \quad I'(x_w) = -\log P'(x_w).$$

*Then the drop in self-information, defined as*

$$\Delta I(x_w) = I'(x_w) - I(x_w),$$

*is bounded by*

$$\Delta I(x_w) \leq -\log\Big(1 + \tfrac{\delta}{P_{\max}}\Big),$$

*where $P_{\max}$ is a small upper bound on $P(x_w)$ in the high self-information region.*

*Proof.* **Step 1: Express $\Delta I(x_w)$.**

$$\Delta I(x_w) = [-\log(P(x_w) + \delta)] - [-\log P(x_w)] = -\log\big(P(x_w) + \delta\big) + \log P(x_w).$$

**Step 2: Normalize by $P(x_w)$.**

$$\Delta I(x_w) = -\log\Big[P(x_w)\big(1 + \tfrac{\delta}{P(x_w)}\big)\Big] + \log P(x_w) = -\log\Big(1 + \tfrac{\delta}{P(x_w)}\Big).$$

**Step 3: Bound using $P_{\max}$.** Since $P(x_w) \leq P_{\max}$ and $P_{\max} \ll 1$, we have

$$1 + \tfrac{\delta}{P(x_w)} \geq 1 + \tfrac{\delta}{P_{\max}}.$$

Hence,

$$-\log\Big(1 + \tfrac{\delta}{P(x_w)}\Big) \leq -\log\Big(1 + \tfrac{\delta}{P_{\max}}\Big),$$

giving

$$\Delta I(x_w) \ \leq \ -\log\!\left(1 + \tfrac{\delta}{P_{\max}}\right).$$

Thus, even after watermarking, a low-probability token remains in a high-self-information regime (downward shift is limited). $\square$

**Lemma H.2** (Concentration in High Self-Information Region)**.** *Consider a watermarking process that selectively increases the probability of certain tokens by $\delta$. Let $I_\alpha$ be the $\alpha$-quantile of self-information values under $M_{attack}$, i.e.,*

$$\Pr_{y_i \sim M_{attack}}\!\big[I(y_i) \geq I_\alpha\big] = \alpha.$$

*For sufficiently small $\delta$, any watermarked token $y_i$ satisfies*

$$I(y_i) \ \geq \ I_\alpha \ - \ C_\delta,$$

*where $C_\delta$ is a small constant capturing the maximum self-information drop due to $\delta$.*

*Proof.* Let $y_i \in \mathcal{W}$ be a watermarked token. Its probability is raised from $P(y_i)$ to $P(y_i) + \delta$. Using Lemma H.1,

$$I(y_i) - I'(y_i) \ \leq \ -\log\!\left(1 + \tfrac{\delta}{P_{\max}}\right) = C_\delta.$$

We denote

$$C_\delta = \sup_x \big|\Delta I(x)\big|.$$

Because tokens subject to watermarking are chosen (pre-watermark) from the high self-information region ($I(y_i) \geq I_\alpha$ except possibly some edge cases), their self-information remains at least $I_\alpha - C_\delta$ after the slight probability increase. Thus,

$$I'(y_i) \ \approx \ I(y_i) \ \geq \ I_\alpha - C_\delta,$$

showing that watermarked tokens are still near or above $I_\alpha - C_\delta$. $\square$

### H.3. Bounds on Attack Success Rate

**Definition H.3** (Attack Success)**.** Let $W_i$ be the event "token $y_i$ is watermarked," and let $A_i$ be the event "token $y_i$ is removed (masked) by the attack." The overall attack is considered *successful* if *every* watermarked token is removed:

$$\text{Success}(y) = \bigwedge_{i \in \mathcal{W}} A_i.$$

Equivalently, $\text{Success}(y)$ requires $I(y_i) \geq \tau_\epsilon$ for all $i \in \mathcal{W}$, where $\tau_\epsilon$ is the chosen self-information threshold (i.e., the $\epsilon$-percentile).

**Theorem H.4** (Attack Success Probability Bounds)**.** *Let $\tau_\epsilon$ be the self-information threshold chosen by the attacker. Then:*

1. ***Lower Bound:***

$$\Pr\big[Success(y)\big] \ \geq \ \left(\min_{i \in \mathcal{W}} \Pr[I(y_i) \ \geq \ \tau_\epsilon \ | \ W_i]\right)^{|\mathcal{W}|}.$$

2. ***Upper Bound:***

$$\Pr\big[Success(y)\big] \ \leq \ \Pr\!\Big[\bigcap_{i \in \mathcal{W}} \{\, I(y_i) \ \geq \ \tau_\epsilon \}\Big].$$

*Proof.* **Step 1: Success Event.** The event $\text{Success}(y)$ is equivalent to

$$\bigcap_{i \in \mathcal{W}} \{I(y_i) \ \geq \ \tau_\epsilon\}.$$

If any watermarked token $y_i$ has $I(y_i) < \tau_\epsilon$, it is not masked and the watermark may remain, so the attack fails.

**Step 2: Lower Bound.** Let

$$\alpha_i = \Pr[I(y_i) \geq \tau_\epsilon \mid W_i].$$

Under (conditional) independence or a suitable lower-bounding assumption,

$$\Pr\Big[\bigcap_{i \in \mathcal{W}} \{I(y_i) \geq \tau_\epsilon\} \ \Big| \ W_i \text{ for each } i\Big] \geq \prod_{i \in \mathcal{W}} \alpha_i.$$

Then,

$$\prod_{i \in \mathcal{W}} \alpha_i \geq \Big(\min_{i \in \mathcal{W}} \alpha_i\Big)^{|\mathcal{W}|}.$$

This implies

$$\Pr[\text{Success}(y)] \geq \Big(\min_{i \in \mathcal{W}} \Pr[I(y_i) \geq \tau_\epsilon \mid W_i]\Big)^{|\mathcal{W}|}.$$

**Step 3: Upper Bound.** Clearly,

$$\Pr[\text{Success}(y)] = \Pr\Big[\bigcap_{i \in \mathcal{W}} \{I(y_i) \geq \tau_\epsilon\}\Big] \leq \Pr\Big[\bigcap_{i \in \mathcal{W}} \{I(y_i) \geq \tau_\epsilon\}\Big].$$

In other words, the event that *all* watermarked tokens exceed $\tau_\epsilon$ (and hence are masked) is the maximum possible success scenario for the attacker.

$\square$

## H.4. Corollary: Optimal Threshold in Randomized Watermarking

**Corollary H.5** (Optimal Threshold under Random Watermarking). *Suppose a watermarking scheme targets tokens above the $\gamma$-quantile $I_\gamma$. If the attacker sets $\tau_\epsilon \approx I_\gamma$, then asymptotically:*

$$\Pr[\text{Success}(y)] \approx (1 - \eta)^{|\mathcal{W}|},$$

*where $\eta$ is a small factor that captures the mismatch in the shifted distribution (i.e., how many tokens originally near $I_\gamma$ drop below $\tau_\epsilon$ after probability adjustments).*

*Proof.* **Step 1: Setup.** Under the random watermarking assumption, tokens are chosen in the top $\gamma$-quantile of self-information (i.e., $I(y_i) \geq I_\gamma$). After adding a small $\delta$, their self-information decreases by at most $C_\delta$ (Lemma H.1).

**Step 2: Define $\eta$.** Let

$$\eta = \Pr\Big[I'(y_i) < \tau_\epsilon \ \Big| \ I(y_i) \geq I_\gamma\Big].$$

If $\tau_\epsilon \approx I_\gamma$ and $\delta$ is small, $\eta$ is small because the shift from $I(y_i)$ to $I'(y_i)$ is minor.

**Step 3: Success Probability.** By Theorem H.4, each watermarked token has at least $(1 - \eta)$ probability of lying above $\tau_\epsilon$, so

$$\Pr[\text{Success}(y)] \geq (1 - \eta)^{|\mathcal{W}|}.$$

Similarly, it cannot exceed the intersection probability that *all* watermarked tokens are above $\tau_\epsilon$, and $(1 - \eta)^{|\mathcal{W}|}$ is a good approximation when $\eta \ll 1$. Thus the success probability is high. $\square$

The lemmas and theorems above show how minimal probability boosts ($\delta$) ensure that watermarked tokens remain in the high self-information region. By selecting a threshold $\tau_\epsilon$ near that region, the attacker can mask or replace the majority of these tokens. Key points include:

- **Local Context Benefits**: Self-information depends on the context, making it more precise than a global entropy measure.

- **Small $\delta$ Requirement**: Watermarking must keep $\delta$ small to avoid degrading output quality, which in turn prevents drastic drops in self-information for chosen tokens.

- **Success Probability Bounds**: Theorem H.4 establishes that success probability can be arbitrarily close to 1 for an appropriate threshold.

- **Near-Optimal Threshold**: Corollary H.5 suggests an attacker should roughly match the watermark's targeted quantile for best results.

The theoretical proof for the self-information rewrite attack shows that by a well-chosen filter we could remove the watermark in the given text effectively. Our lemmas and bounds show that, under small-$\delta$ constraints, high-entropy tokens remain detectable as high self-information tokens. This explains why token-level self-information filtering outperforms global, context-agnostic entropy filtering. Consequently, an attacker can remove most or all watermarked tokens while maintaining strong semantic coherence in the final paraphrased text.

## I. Visualization

In this section, we present a visual comparison of our algorithm with other model-based paraphrasing methods, along with the corresponding z-scores after the attack. For discrete methods, green tokens are marked in green, and red tokens in red. In the watermarking algorithm, the detector identifies the embedded watermark through green tokens and calculates the z-score; fewer green tokens or a lower z-score indicate a more successful attack. For continuous methods, the shade of color denotes the weight of the watermarked token, with darker colors representing higher weights. In the case of attacked text, lighter colors indicate a more successful attack.

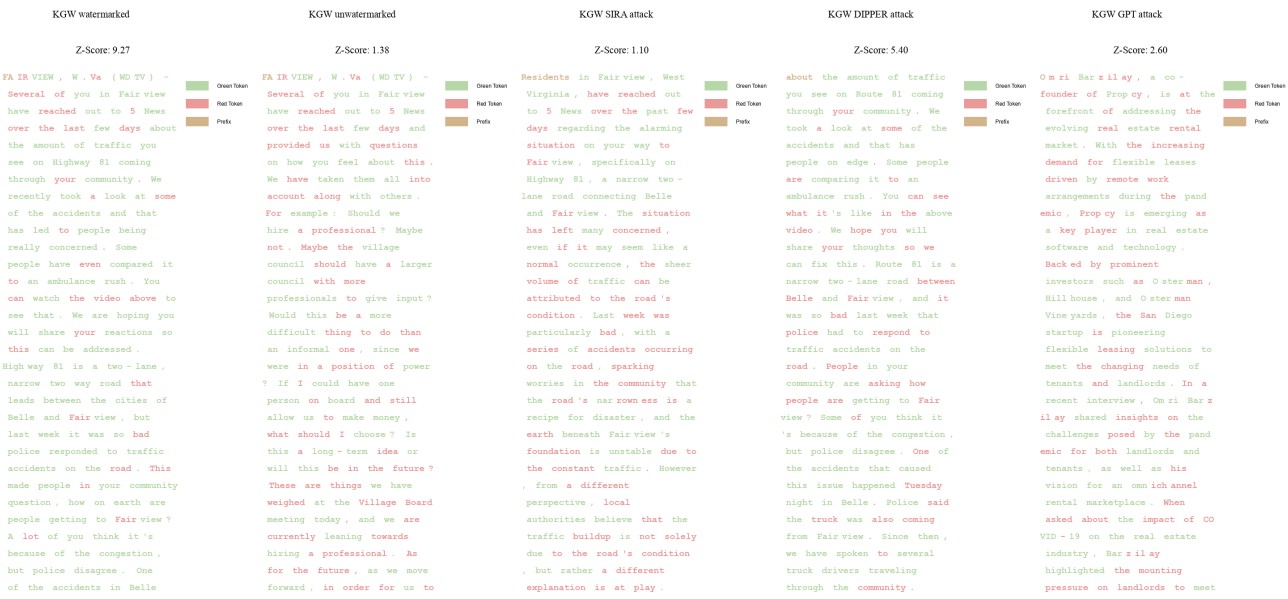

*Figure 4.* Comparison of different paraphrasing methods on KGW watermarks. Each word's color indicates whether it is a green or red token. **Fewer green words/lower z-scores** suggest a more effective paraphrasing approach. The unwatermarked text represents the model's output without the influence of the watermarking algorithm. The example demonstrates that our method achieves a better z-score than the unwatermarked text..

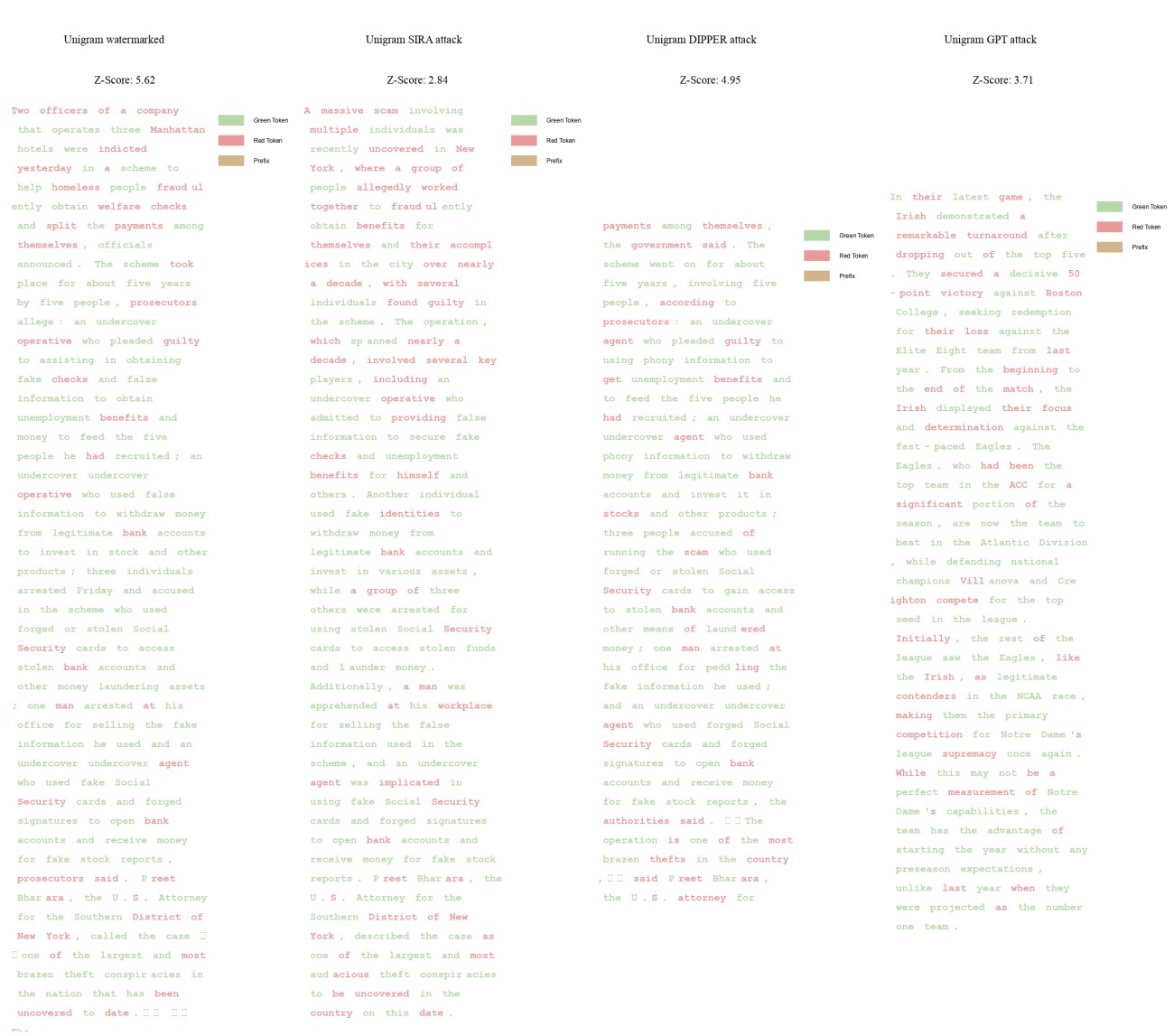

*Figure 5.* Comparison of different paraphrasing methods on Unigram watermarks.

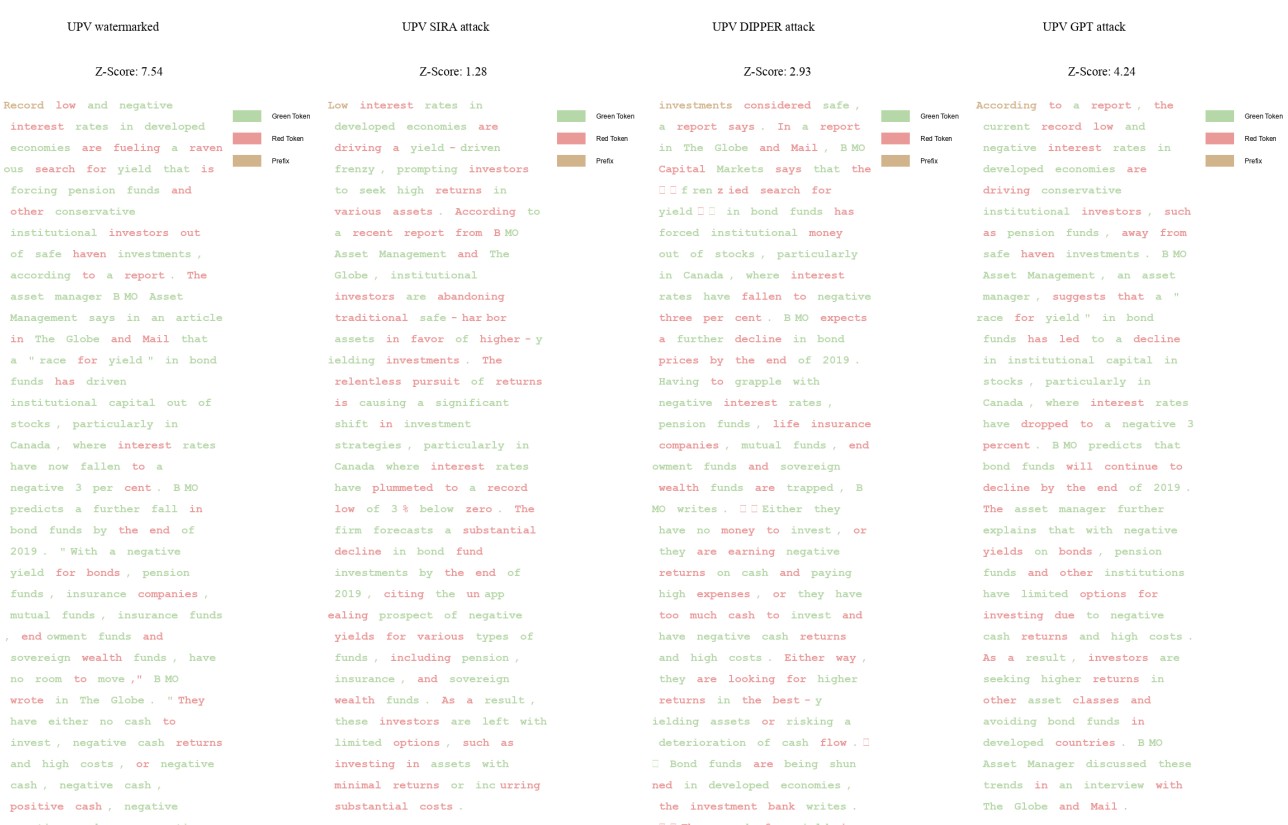

Figure 6. Comparison of different paraphrasing methods on UPV watermarks. The color of each word indicates whether it belongs to a green token or a red token. **Less green** signifies a more effective paraphrasing approach. Our methods show better performance in removing original watermark text green token.

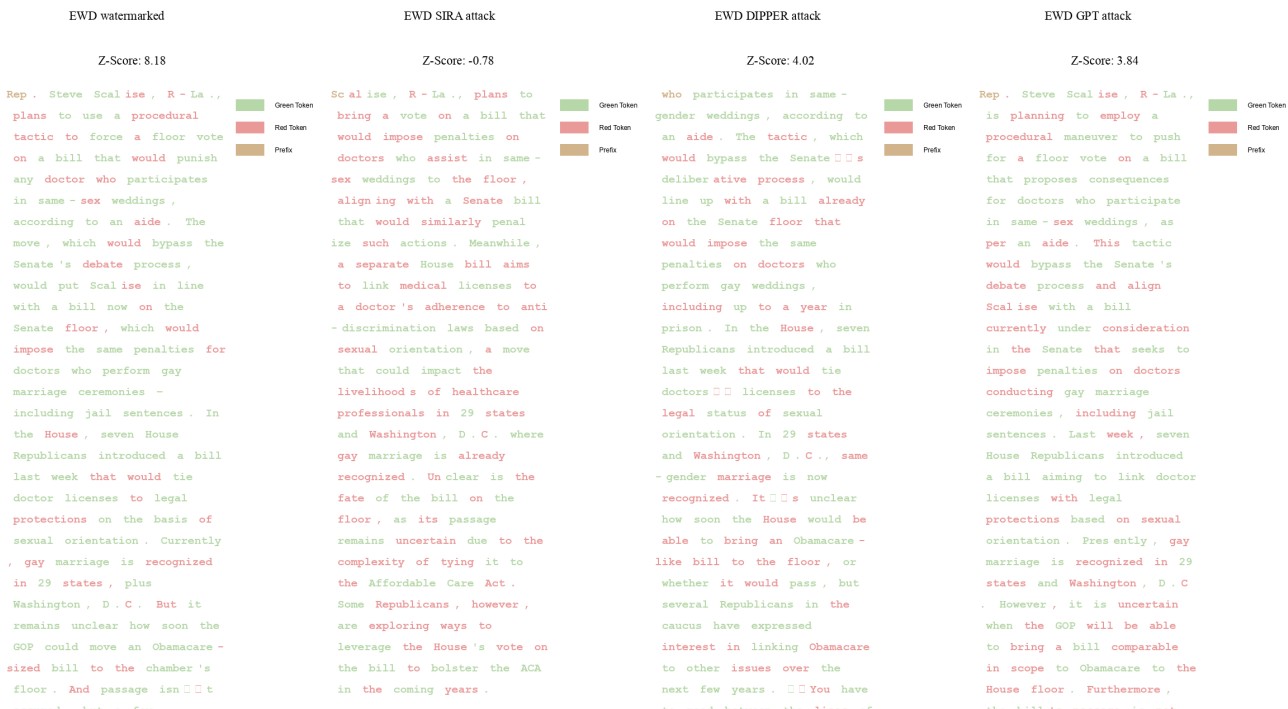

Figure 7. Comparison of different paraphrasing methods on EWD watermarks.

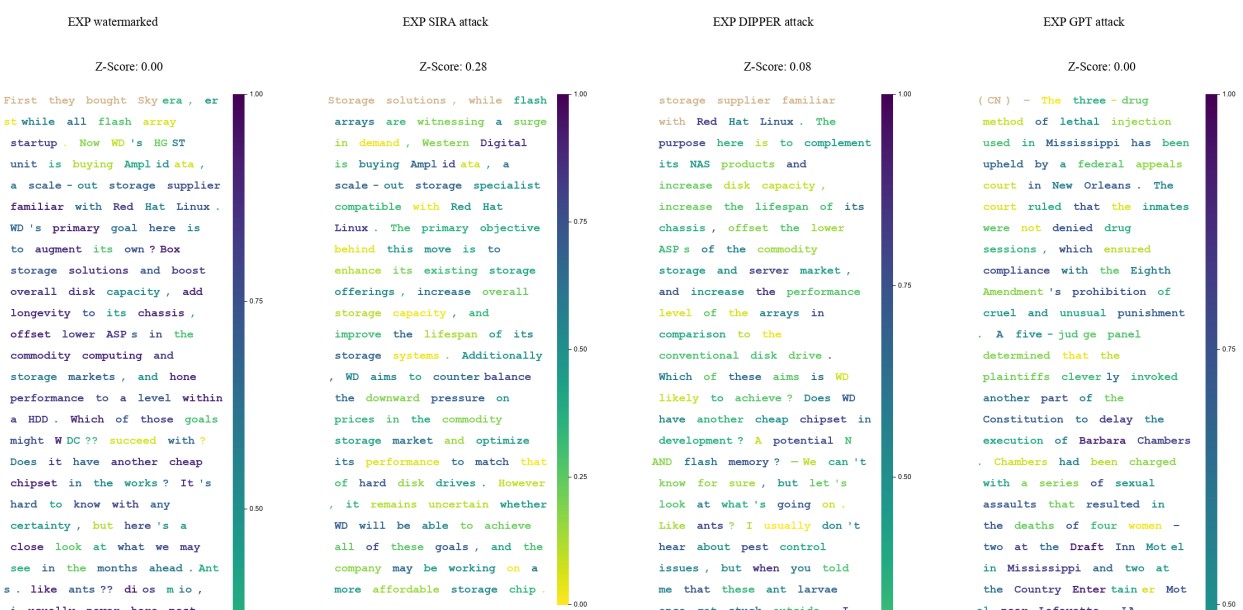

*Figure 8.* Comparison of different paraphrasing methods on EXP watermarks. The color of each word indicates whether it is a green or red token. For EXP, **lighter word colors and higher z-scores indicate a more effective attack**.

*Table 9.* Comparison of watermark algorithms under different attack methods on the OpenGen dataset. Our proposed methods SIRA-Tiny and SIRA-Small outperform all previous paraphrasing-based attacks under black-box settings.

| | Attack Success Rate (%) on OpenGen Dataset | | | | | | |
|---|---|---|---|---|---|---|---|
| **Attack** / Watermark | **KGW-1** | **Unigram** | **UPV** | **EWD** | **DIP** | **SIR** | **EXP** |
| **Word delete** | 21.8 | 0.8 | 9.6 | 17.6 | 64 | 36.8 | 6.6 |
| **Synonym** | 77.4 | 16.8 | 67.8 | 72 | 98 | 71.4 | 47.4 |
| **GPT** | 69 | 58.2 | 57.4 | 73.4 | 98.2 | 58.8 | 74.2 |
| **DIPPER-1** | 89.4 | 67.8 | 71.4 | 88.8 | 98.8 | 74.6 | 83.2 |
| **DIPPER-2** | 89.2 | 71.2 | 78.8 | 92.2 | 99.0 | 72.8 | 85.6 |
| **SIRA-Tiny (Ours)** | 92 | 84 | 74.8 | 94.2 | 99.6 | 74.6 | 81.8 |
| **SIRA-Small (Ours)** | 93.8 | 91.2 | 80.6 | 94.8 | 99.6 | 80.2 | 86.2 |

## J. Extend Experiments

We conduct additional experiments on the OpenGen dataset (Krishna et al., 2024), which consists of sampled passages from WikiText-103. Specifically, we use a subset of 500 chunks as prompts, following the same experimental protocol described in our main evaluation—namely, we prompt a target LLM with each chunk and assess the ability of different watermark removal methods to induce decoding failures in the watermark verifier. We report the attack success rate (ASR) as our primary metric. As shown in Table 9, our proposed methods consistently achieve the highest ASR across all watermarking algorithms, demonstrating strong generalization and robustness beyond the training or development set used in previous sections.

We observe that the performance of DIPPER improves significantly on the OpenGen dataset compared to its performance on C4. We hypothesize that this may be attributed to distributional similarity between OpenGen and the supervised training data used to train the DIPPER paraphraser, as both originate from the same source corpus introduced in Krishna et al. (2024). Despite this advantage, our proposed method SIRA-Small still consistently achieves the highest attack success rates across all watermarking algorithms, demonstrating stronger generalization to diverse data distributions.

*Table 10.* Attack success rate (ASR) on Adaptive Watermark and Waterfall Watermark. Sample size = 200. Our methods outperform all baselines across both settings.

| Adaptive Watermark | | Waterfall Watermark | |
|---|---|---|---|
| **Attack** | **ASR (%)** | **Attack** | **ASR (%)** |
| Word delete | 5.6 | Del | 4.4 |
| Synonym | 92.4 | Syn | 55.6 |
| GPT-4o Paraphraser | 61.4 | GPT | 80.0 |
| DIPPER-1 | 60.6 | DIPPER-1 | 73.8 |
| DIPPER-2 | 65.6 | DIPPER-2 | 80.0 |
| SIRA-Tiny (Ours) | 96.2 | SIRA-T (Ours) | 88.4 |
| SIRA-Small (Ours) | **98.2** | SIRA-S (Ours) | **90.8** |

To further evaluate the generalizability of our proposed methods, we conduct additional experiments on two recently introduced watermarking schemes: Adaptive Watermark (Liu & Bu, 2024) and Waterfall Watermark (Lau et al., 2024). Following the same black-box threat model, we apply both baseline and our proposed attack methods to 200 samples from C4 dataset for each setting. As shown in Table 10, SIRA-Tiny and SIRA-Small achieve significantly higher attack success rates (ASR) compared to all baselines, including the strong GPT-4o paraphraser and DIPPER variants. Notably, on Adaptive Watermark, SIRA-Small reaches 98.2% ASR, while the strongest baseline only achieves 92.4%. Similarly, for Waterfall Watermark, SIRA-Small obtains 90.8% ASR, outperforming the closest baseline by over 10 percentage points. These results demonstrate the superior robustness and transferability of our attack methods across different watermarking designs.

