# OpenReview forum: "Revealing Weaknesses in Text Watermarking Through Self-Information Rewrite Attacks"
_ICML.cc/2025/Conference — ICML 2025 poster_

### Official Review · Reviewer_7XjW · 2025-03-12

**Overall Recommendation:** 2

**Summary:**

The paper proposes a new attack against model-watermarking algorithms that involves first identifying tokens in an LLM output that have high self-information, and then passing the output to a paraphraser that changes these tokens. The hypothesis is that these tokens are also the tokens that most likely contain the watermark signals, hence such targeted paraphrasing could improve the efficiency of paraphrasing attacks against model watermarking methods. The paper presented empirical results to support their claims, showing that the proposed method can achieve much higher attack success rates compared to benchmarks, at a relatively low per token cost.

**Claims And Evidence:**

The claim that their method outperforms baselines empirically is partially supported by the existing experiments, though there are some additional results that would make them more convincing.
- The trade-off between semantic preservation of the attack and the effectiveness of the attack. Currently this is analyzed only in aggregate in separate charts. A Pareto-plot showing the trade-off between attack effectiveness and semantic preservation (e.g. measured by semantic similarity) across benchmarks (rather than just on the proposed method) will help.
- Related point: the use of \epsilon is confusing, as it denotes three different things in the paper, i.e., in equation 4 for semantic similarity threshold, in unnumbered equation line 269 for percentile threshold, and line 212 for watermarking threshold.
- Inclusion of error bars will also confirm that the results are statistically significant.
- If the claim is that self-information is the best metric to identify tokens with watermark signal, the best approach to analyze this is to directly compute how well the proposed metric predicts which are the tokens that contain the 'greenlist' signals, e.g. by computing some appropriate correlation metric.
- Smaller point: it would be useful to see the naive baseline results of a paraphrasing attack using the same base paraphraser used in the SIRA methods, i.e. unconstrained paraphrasing (with similar instructions to not reuse the same words in the reference text), to see the direct impact of the identification of a subset of tokens for replacement.

The bigger problem lies in the proposed explanation and theoretical analysis.
- Entropy is expected self-information, and it is not true that "higher entropy is typically associated with high self-information". Choosing higher self-information tokens based on the proposed method essentially means choosing tokens that have lower absolute probabilities given preceding tokens. Possible explanations for the performance gains may be considered more from this perspective.
- The proposed detailed discussion in Appendix G/H is unclear that needs to be a lot more detailed and careful with assumptions and approximations made. Statements like line 1041 on "entropy is context-agnostic" needs to be properly justified and defined.
- The authors should be a lot clearer what is the experimental setting and what they specifically did to compute the results for Table 9.
- The problem formulation in Sec 3.1 does not directly relate to the crux of the proposed method, which is token-level based. The method is broadly on identifying tokens to mask and replace, while the formulation do not related watermarking methods to token-level perturbation (definition 3).

**Essential References Not Discussed:**

As mentioned, it would be useful to discuss logit perturbation-based watermarking approaches that go beyond just 'green-red list' approaches (e.g., [1] mentioned above and related works). As the proposed method is primarily designed based on the 'green-red list' watermarking model, discussing other approaches that spread the watermarking signal across all tokens would better illustrate the generalizability, or limitations, of the proposed method.

**Experimental Designs Or Analyses:**

Please see issues above on metrics and datasets.

Additionally, it would be additional validation for the authors to also consider recent robust watermarking methods such as [1], rather than just logits adjustment-based model watermarking methods that adopts 'green-red' list approaches (which the proposed method is deigned directly to attack). For e.g., [1] applies logits perturbations to the entire token space but with varying degree of perturbation to each token based on a hash, preceding tokens and chosen perturbation functions.

Showing that the proposed method can also work for that and generalize beyond just 'green-red list' approaches, will significantly strengthen the validity of the method.

[1] Lau et al, Waterfall: Scalable Framework for Robust Text Watermarking and Provenance for LLMs, EMNLP 2024

**Methods And Evaluation Criteria:**

Please see above point and question regarding direct analysis of the correlation and prediction capabilities of the self-information metric v.s. the 'watermark signal' tokens.

It would also be better if experiments are conducted on more than just one dataset, i.e. the C4 dataset, especially since the proposed method should be relatively easy to implement for other datasets. The other datasets should ideally cover a different type of text, e.g. not news articles related but perhaps more scientific or literature-based, to confirm that the underlying token distribution of the dataset do not significantly affect the performance of the method.

**Other Comments Or Suggestions:**

Please see above

**Other Strengths And Weaknesses:**

Overall, I think this has potential to be a good contribution to the literature if the attack is more rigorously backed empirically. The theoretical underpinnings and explanation is problematic, but given that this is a proposed attack model and is primarily empirical in nature,  it may be ok as long as these portions are clarified and de-emphasized in the paper.

**Questions For Authors:**

Please see the points above, on empirical gaps and questions on explanations regarding why the proposed method works.

**Relation To Broader Scientific Literature:**

Empirically, it seems like there is merits to consider this attack in future watermarking works, if the authors can more rigorously justify the proposed method's performance gains compared to existing paraphrasing attacks. It would be useful for the authors to also discuss or even better evaluate other logits perturbation-based model watermarking approaches that go beyond the 'green-red list', such as [1] mentioned above and other similar types of approaches.

**Theoretical Claims:**

Please see above regarding the issues on self-information vs entropy, problem formulation gap on relating token level watermarking to identification of 'watermarked tokens' as a viable attack strategy. Conceptually, basic watermarking methods operating on the 'green-red' list approach also relies on both the green and red lists for signal detection and watermarking strength -- so it is unclear how 'green tokens' identification is more important than 'red tokens', rather than the actual distortion from the underlying word distribution.

---

> ### Author Rebuttal · Authors · 2025-03-31
>
> We are thankful to Reviewer 7XjW for the thorough and detailed feedback, due to space limit, we address the main concerns below:
>
> > Q1: A Pareto plot, the use of \epsilon, Line 1041
>
> A1: We greatly appreciate the reviewer’s suggestion. We will add the Pareto plot, correct the misuse of \epsilon, and revise Line 1041 to avoid confusion.
>
> >Q2: Directly computing how well the proposed metric predicts tokens will be helpful
>
> A2: Due to space limitations, please see the reply to zCjb A1.
>
> >Q3: It would be useful to see the naive baseline results of a paraphrasing attack.
>
> A3: We clarify that this is already included in our paper. As shown in Table 3, we compare our method to a naive baseline where the LLM paraphrases the input twice. This aligns with our approach, where contains two paraphrasing for reference text and attack text. Results show that our design outperforms naive baseline.
>
> >Q4: Possible explanations for the empirical and theoretical
>
> A4: We clarify that prior works claim to embed watermarks in high-entropy is a heuristic description, **they do not explicitly compute token-level entropy to build the greenlist**. Instead, greenlist tokens are selected via a hash-based pseudorandom process conditioned on past tokens and a key, implicitly simulating high-entropy selection.
>
> For example, in EWD [3], The high entropy concept is represent as a renormalization step:
>
> ```python
> renormed_probs = probs / (1 + z_value * probs)
> sum_renormed_probs = renormed_probs.sum(dim=-1)
> ```
>
> This renormalization acts as soft reweighting that suppresses high-probability tokens and flattens the distribution, thereby emphasizing low-probability regions. As a result, low-probability tokens are further highlighted and tend to receive higher weights regarding the watermark signals. Similarly, EXP uses exponential sampling to favor low-probability tokens, which, once selected, have higher weight during detection.
>
> In summary,**low-probability tokens are either preferred when selecting a green token or assigned stronger watermark weights once selected**. Our self-information approach aligns more directly with practical implementation, enabling more accurate identification of watermark-bearing tokens and improving attack effectiveness.
>
> > Q5: The formulation in Sec 3.1 doesn't connect watermarking to token-level perturbations .
>
> A5: We clarify that the perturbation function in Section 3.1 refers to the LLM used for paraphrasing. Definition 3 is **not** meant to describe token-level watermark perturbations. It corresponds to the second step of our method, where the LLM is instructed to paraphrase the input.
>
> >Q6:  Why is the green token more important?
>
> A6: The watermark signal is embedded by biasing generation toward green tokens, while red tokens act as a control group. Detection relies on whether the green token significantly exceeds the non-watermarked baseline. Thus, the green tokens carry the actual signal. Regarding distortion, robust watermarking schemes are explicitly designed to minimize it while preserving detectability; otherwise will affect text quality and stealth.
>
>
> >Q7: Other dataset experiment
>
> A7: We follow the reviewer’s suggestion add the OpenGen dataset [2], which samples from WikiText-103. We use the 500 chunks as prompts, following the same evaluation protocol as in our main experiments. We show our results below, and our methods achieve the highest ASR.
>
> | Attack  | KGW | Uni | UPV | EWD | DIP | SIR | EXP |
> |---------|-----|-----|-----|-----|-----|-----|-----|
> | Del     |21.8|0.8 |9.6 |17.6|64  |36.8|6.6 |
> | Syn     |77.4|16.8|67.8|72  |98  |71.4|47.4|
> | GPT     |69  |58.2|57.4|73.4|98.2|58  |74.2|
> | DIPPER-1   |89.4|67.8|71.4|88.8|98.8|74.6|83.2|
> | DIPPER-2   |89.2|71.2|78.8|92.2|99.0|72.8|85.6|
> | SIRA-T    |92  |84  |74.8|94.2|99.6|74.6|81.8|
> | SIRA-S   |93.8|91.2|80.6|94.8|99.6|80.2|86.2|
>
>
>
>
>
> >Q8: Waterfall watermark experiment
>
> A8: We followed the reviewer’s suggestion to conduct the experiment on the waterfall watermark[1]. We use 500 samples from the C4 dataset and select 0.2 as our z-threshold; others follow the same evaluation protocol as in our main experiments. The results shown below, our methods get the highest ASR.
>
> | Waterfall          | ASR    |
> |--------------------|--------|
> | Del        | 4.4   |
> | Syn             | 55.6  |
> | GPT | 80    |
> | DIPPER-1           | 73.8  |
> | DIPPER-2           | 80    |
> | SIRA-T          | 88.4  |
> | SIRA-S         | 90.8  |
>
> >Q9: Weaknesses regarding theoretical
>
> A9: We thank the reviewer for the constructive suggestion. We acknowledge that our work is primarily empirical, and while we attempted to provide theoretical insights, some parts may be limited or unclear. We will revise the manuscript accordingly and de-emphasize those sections as suggested. We appreciate any further feedback to improve the theoretical part.
>
> ---
>
> Anonymous Reference Link: https://docs.google.com/document/d/1-VCOEO5eJmrq-_44oGaHdRE7KOGXHumV75h9DDCTP8A/edit?usp=sharing

---

> > ### Comment · Reviewer_7XjW · 2025-04-04
> >
> > Thanks for the response. While some of my concerns have been addressed such as the experiments with an additional dataset and with different watermarking approach, there are still issues unaddressed.
> >
> > As mentioned before, the Pareto plot would help characterize the trade-off between semantic preservation of the attack and the effectiveness of the attack.
> >
> > Inclusion of error bars will also confirm that the results are statistically significant.
> >
> > Correlation between score and greenlist signal tokens.
> >
> > The naive baseline results of a paraphrasing attack using the same base paraphraser used in the SIRA methods, i.e. unconstrained paraphrasing (with similar instructions to not reuse the same words in the reference text), to see the direct impact of the identification of a subset of tokens for replacement — the current paraphrasing attacks seems to not use the same instructions as the method, to not use the same words in the reference text etc.
> >
> >
> > Hence, I will maintain my score.

---

> > > ### Author Response · Authors · 2025-04-04
> > >
> > > We thank Reviewer 7XjW for the additional feedback. Due to character limits and the number of initial comments, we are unable to address every detail. We further respond to key concerns below:
> > >
> > > > **Q1: Pareto plot**
> > >
> > > **A1:** As stated in our previous response, we will add the Pareto plot in the revised version to help visualize trade-offs. This addition supports but does not affect our core contributions.
> > >
> > > > **Q2: Error bars**
> > >
> > > **A2:** Thanks for the new suggestion. We will include error bars in the revised manuscript to clarify variability.
> > >
> > > > **Q3: Correlation between detection score and greenlist tokens**
> > >
> > > **A3:** We would like to clarify that due to space constraints, we were unable to provide a detailed explanation of the correlation between the detection score and greenlist signal tokens in the initial rebuttal.
> > >
> > > The detection z-score is explicitly designed to reflect how many greenlist tokens remain in the generated text. These tokens are selected during generation through a hash-based pseudorandom process that conditions on the past tokens and a secret key, and are favored via a bias added to the logits.
> > >
> > > In the naive watermarking method [1], under the null hypothesis, the expected number of green tokens is $\gamma \cdot T$, with $\gamma$ as greenlist ratio and $T$ the token count. The z-score is:
> > > $$
> > > z = \frac{G - \gamma T}{\sqrt{T \cdot \gamma (1 - \gamma)}}
> > > $$
> > > where $G$ is the number of tokens in the generated text that fall into the greenlist at their corresponding positions. This statistic measures how much the observed green token count deviates from the expected value under randomness. Since watermarked generation increases the probability of sampling green tokens, $G$ tends to be significantly higher in watermarked texts, leading to a large positive z-score. Therefore, there is a direct and quantifiable correlation between the detection score and the number of greenlist tokens preserved in the text.
> > >
> > >
> > > Moreover, simply checking how many high self-information tokens fall into the greenlist is unreliable. Token falls in green token may carry different weights. For example, in EWD [2], high-information green tokens have much larger impact than low-weight ones. Thus, the detection score reflects both quantity and quality of green tokens, making simple matching strategies suboptimal.
> > >
> > > > **Q4: Instructions for paraphrasing**
> > >
> > > **A4:** We respectfully argue that using different instructions is a common and accepted practice in prior work[3,4], and in our case, it is inherently part of the method to adapt the algorithm design. Different methods are designed with different assumptions and mechanisms, and their corresponding instructions are often not interchangeable.
> > >
> > > For example, DIPPER [3]  **needs to preprocess input text and inject customized prompts like** `"lexical = {lex}, order = {order}, {curr_sent_window}"` to give the model a hint and adapt raw text to how it was trained. Similarly, Sadasivan et al. [4], the authors **explicitly define a customized instruction (Appendix B.2)**, stating that the output should be as **diverse and different as possible from the input**, and **should not copy any part verbatim**.
> > >
> > > Our novel mask-and-paraphrase framework requires its own instruction design, making this component integral to the method. We kindly ask the reviewer to reconsider this concern, **given the methodological differences and the established practices in prior work**.
> > >
> > > In addition,  **we believe we have addressed the core concerns raised in the initial review** —including an additional dataset (A7), a new watermarking baseline (A8), and the rationale behind low-probability tokens (A4)—which further supports the soundness of our approach.
> > >
> > > **We would sincerely appreciate it if you could consider increasing your score to reflect the revised manuscript.**
> > >
> > > **References**
> > > [1] Kirchenbauer, John, et al. "A watermark for large language models." . ICML(2023)
> > >
> > > [2] Lu, Yijian, et al. "An entropy-based text watermarking detection method." *arXiv preprint arXiv:2403.13485* (2024).
> > >
> > > [3] Krishna, Kalpesh, et al. "Paraphrasing evades detectors of AI-generated text, but retrieval is an effective defense." NeruIPS(2023).
> > >
> > > [4] Sadasivan, Vinu Sankar, et al. "Can AI-generated text be reliably detected?." arXiv preprint (2023).

---

### Official Review · Reviewer_P6b2 · 2025-03-13

**Overall Recommendation:** 4

**Summary:**

The paper aims to erase text watermarks from LLMs by proposing a novel rewrite attack utilizing self information. The proposed SIRA could achieve almost 100% success rate across various watermark algorithms. Specifically, SIRA calcuates self information of every token in the watermarked sequence, where tokens of high values are masked in the output. To complete the masked sequence that includes all information from the original sequence, a paraphrased version of the original sequence will be used as a reference. Experiments show that the text quality is well preserved, while maintaining fine complexity and budget.

**Claims And Evidence:**

It seems the text quality, while superior than most baselines, is slightly lower than the GPT paraphrasing. In the paper, you claimed that your method has smaller impact on text quality than other baselines.

**Essential References Not Discussed:**

No

**Experimental Designs Or Analyses:**

No issues.

**Methods And Evaluation Criteria:**

Since the method is completely black-box, which indicates the user prompt when generating the watermarked sequence is not available as wee, does this missing prompt affect the calculation of self-information? Also, if the given watermarked texts is cropped from a long output, will the missing context affect the calculation of self-information?
I wonder if the parapharsed reference text is necessary as maybe during the masked sequence completion, the llama model could be instructed to avoid using exactly same wording from the original text. If this step is unnecessary as you use the original text as reference, the cost and time consumption could be further lowered.

**Other Comments Or Suggestions:**

No issues.

**Other Strengths And Weaknesses:**

It is nice that proposed method is a plug-and-play & black-box attach method while many previous works have more assumption on the knowledge of watermarking generation process.

**Questions For Authors:**

Please refer to previous sections.

**Relation To Broader Scientific Literature:**

No issues.

**Theoretical Claims:**

No issues.

---

> ### Author Rebuttal · Authors · 2025-03-31
>
> We are thankful to the reviewer **P6b2** for the appreciation of our work and the efforts spent to review our paper. We address concerns and questions below:
>
> > **Q1: Text quality slighter lower than GPT paraphrase**
>
> **A1:** We thank the reviewer for their suggestion and will revise the description of our experimental conclusions accordingly. We fully agree with the reviewer and would like to clarify that text quality is largely influenced by the choice of **paraphrasing model**. In our case, we adopted GPT-4o; the inherent ability difference of LLMs causes such a gap. For the stronger LLM like LLaMA3-70B, it outperforms GPT in two watermarks. One advantage of our method is its transferability: unlike DIPPER, our approach is training-free. As more powerful LLMs become available in the future, our method can be seamlessly adapted with zero cost to take advantage of better text quality.
>
> > **Q2: Does this missing prompt affect the calculation of self-information?**
>
> **A2:** This is a great question. Our answer is no—the calculation of self-information does not need such a prompt. The prompts used to generate watermarked text in prior works such as DIPPER [1] and Random Walk [2] are typically referred to as “context.” However, our goal is to minimize assumptions in order to make the method a simple and broadly applicable tool. In the results we present, we do not use any such context prompts. In earlier studies, the original prompts were primarily designed to help paraphrasing models better preserve semantics through tailored instructions. We consider exploring the impact of prompt design on our method as an interesting direction for future work.
>
> > **Q3: If the given watermarked text is cropped from a long output, will the missing context affect the calculation of self-information?**
>
> **A3:** Yes, but it depends on how long the cropped text is. The missing context theoretically affects how self-information is calculated since it depends on the exact preceding context. However, our watermarking approach remains effective in practice because it primarily relies on local context within the segment, rather than requiring the entire text. Consequently, cropping a reasonable portion of the output does not significantly degrade our detection or performance unless the given text is too short or the different segment contexts are unrelated. In such an extreme case, our method degrades to a performance level similar to that of random masking.
>
> > **Q4: Using instruction instead of reference text**
>
> **A4:** We thank the reviewer for the suggestion. This was indeed one of the approaches we explored during the preliminary phase of our study. Adding explicit instructions to encourage the LLM to use different words proved to be ineffective; a large performance gap will exist, especially for lightweight models such as those at the 3B scale. We believe this is because such a small model cannot fully understand complex instructions. While this approach performs comparably on larger models (e.g., 70B), it still underperforms compared to using a reference text. This is likely because, as mentioned in lines 256–260, we instruct the model to do tasks similar to filling in blanks, and due to the high similarity between mask text and watermarked text, LLMs tend to take shortcuts by copying directly from the original text; instructions are not sufficient to fully prevent this behavior. We will continue to explore whether better prompt design can help to achieve this goal.
>
> ---
>
> **References**
>
> [1] Krishna, Kalpesh, et al. "Paraphrasing evades detectors of AI-generated text, but retrieval is an effective defense." *Advances in Neural Information Processing Systems* 36 (2023): 27469–27500.
> [2] Zhang, Hanlin, et al. "Watermarks in the sand: Impossibility of strong watermarking for generative models." *arXiv preprint arXiv:2311.04378* (2023).

---

> > ### Comment · Reviewer_P6b2 · 2025-04-02
> >
> > Thank you for your response to my concerns. I will maintain my rating.

---

> > > ### Author Response · Authors · 2025-04-03
> > >
> > > Dear Reviewer P6b2,
> > > We are truly delighted by your recognition of our work and your interest in our paper. Your feedback is invaluable for enhancing the quality of our manuscript. Many thanks！
> > >
> > > Best regards,
> > >
> > > The Authors

---

### Official Review · Reviewer_zCjb · 2025-03-15

**Overall Recommendation:** 3

**Summary:**

This paper studies how to remove the watermark in text generated by LLMs. It assumes the watermark is injected through the high-entropy (self-information) words. There it first uses an auxiliary mode to compute the self-information for each token in the generated text. Then it masks out the tokens with high self-information and fills in the masks according to the paraphrased text. Experiments show it outperforms baselines.

**Claims And Evidence:**

1. The authors assume existing watermark methods embed patterns in high-entropy tokens and thus base their method on this assumption. However, it's unclear if this is true because there is no theoretical analysis or direct experimental evidence. For example, the authors can add an experiment to directly verify the precision/recall of the masks generated in the first step with the ground truth tokens.
2. Similarly, the authors claim it's the first targeted attack. This "targeted" feature should be clearly defined and validated.
3. There is still a gap between the motivation and the theory. From the beginning, the authors always emphasize that existing watermarks use high-entropy tokens. However, later on, the authors analyze the theory and empirically show that self-information is a better indicator. However, from the motivation, it seems high entropy should give better results than self-information.

**Essential References Not Discussed:**

No

**Experimental Designs Or Analyses:**

1. It's not reasonable that the attackers use much larger models, such as Llama3-3B-70B, to attack watermark text generated by a small model Opt-1.3B. In this case, the attackers can directly use the large models. So a more reasonable attack scenario is the attackers only have limited resources so they can only have a small model, while they want to leverage the capability of a larger model. However, the larger model has watermarks. So they want to use the small model to remove the watermark text generated by the large model. Therefore, it's suggested that similar rules be followed in the evaluation.
2. As mentioned earlier, it would be better to evaluate the accuracy of the detected watermark tokens. Because this is one of the contributions claimed in this paper.
3. It's suggested that the authors evaluate or at least discuss the adaptive watermarks. For example, the watermarks that try to leverage other tokens in addition to the high-entropy loss.

**Methods And Evaluation Criteria:**

make sense.

**Other Comments Or Suggestions:**

Some symbols are used without introduction, such as $Y_w$ at Line 191.

Typo:
Line 1030: "the green tokenthe"

**Other Strengths And Weaknesses:**

Strengths:

1. An interesting idea and experiments show it has high ASRs.
2. Overall, it's easy to follow.


Weakness:

1. It's unclear why authors have several definitions in Section 3.1, especially equations 2-4. Because most of them are not used in the following discussion or design. Why can the watermark defenders see the attack text $y_p$ when they design the detector at Line 192?

2. The figures in the evaluation section are not very visible.

**Questions For Authors:**

Please refer to the above points.

**Relation To Broader Scientific Literature:**

Help people design stronger watermark techniques.

**Theoretical Claims:**

Line 997, why can one assume the model predicts the next words with equal probability? Is it reasonable? Please justify it.

---

> ### Author Rebuttal · Authors · 2025-03-31
>
> We are thankful to the reviewer zCjb for the time spent reviewing our paper. Due to the reply character limit, we address the main concerns below and put the reference in anonymous link:
> > Q1: No theoretical analysis or direct experimental evidence on the high-entropy assumption
>
> A1: The use of high-entropy embeddings is not an assumption but a well-established design choice in prior watermarking research [1–4, 6, 7]. Section 3 of the KGW paper [1] provides a detailed theoretical framework. Notably, we are the first to identify this design as a potential vulnerability exploitable by attackers.
>
> To further support our results, we extend the ablation study in Table 4 with mask text and report the z-score, which reflects the green token remnant.
>
> | Text               | Attack Success Rate | Average *z*-score |
> |--------------------|---------------------|-------------------|
> | Human-written Text | N/A                 | 0.12              |
> | Reference Text     | 64%                 | 3.75              |
> | Attack Text        | 94%                 | 1.85              |
> | Mask Text|100%| 0.66|
>
> > Q2:  Clarification regarding “targeted” attack
>
> A2: We clarify that prior paraphrasing attacks, such as DIPPER [5] and GPT-Paraphraser, rely entirely on the LLM to rewrite text without control over which parts are modified, making them untargeted. In contrast, our method treats watermark removal as a targeted problem, selectively rewriting tokens likely to carry the watermark signal. Experiments show this targeted strategy is more effective.
>
> > Q3:  The motivation gap.
>
> A3:  We clarify that the use of “high entropy” in prior work is heuristic. Specifically, none of the existing watermarking methods explicitly compute entropy, making it reasonable to explore metrics under the heuristic. Entropy and self-information are mathematically related, we find that self-information provides a more direct, token-level signal that better aligns with the actual greenlist selection process. Therefore, our motivation and implementation are consistent: we empirically identify a metric that captures the core vulnerability and reflects the intended intuition.
>
> > Q4: Why can one assume the model predicts the next words with equal probability in Appendix G?
>
> A4: We clarify that this is a deliberate simplification of high entropy and our formal proof, Appendix H does not build on such an assumption. The uniform distribution serves as a theoretical upper bound for high entropy for a given probability space and helps illustrate high-entropy scenarios where probability mass is thinly spread. In such cases, even small probability changes can significantly affect self-information. Thus, this approximation highlights an extreme, simplified case to clarify the scaling behavior under high entropy for the reader not familiar with watermarking.
>
> > Q5: It's not reasonable that the attackers use much larger models.
>
> A5: Our work is aligned with established settings in prior related studies [5–9]. We emphasize that watermark robustness mainly depends on the algorithm design and hyperparameters, not the choice of generation LLM. The mentioned work adaptive watermark in [4] uses the same evaluation. Notably, our method significantly reduces computational and time costs, as shown in Appendix B. We would appreciate references that follow the reviewer's proposed setting regarding LLM watermark robustness evaluation.
>
> > Q6: Evaluate the accuracy of the detected watermark tokens
>
> A6: Please see A1 response.
>
> >Q7: Adaptive watermarks experiments
>
> A7: We appreciate the reviewer’s suggestion. We show the proposed results below; the experiment follows our main experiment setting, and the threshold is 0.75. We report the attack success rate. **As explicitly mentioned in the abstract of the work[4], this watermark is embedded in high-entropy text.** We would appreciate it if the reviewer could clarify what “high-entropy loss” is and could provide a specific reference that did not adopt high-entropy embedding, and we will try our best to include the corresponding experiments during the discussion window.
>
> | Adaptive Watermark  | ASR    |
> |---------------------|--------|
> | Word delete         | 5.6%   |
> | Synonm              | 92.4%  |
> | GPT-4o Paraphraser  | 61.4%  |
> | DIPPER-1            | 60.6%  |
> | DIPPER-2            | 65.6%  |
> | SIRA-Tiny           | 96.2%  |
> | SIRA-Small          | 98.2%  |
>
>
> > Q9: Why can the watermark defenders see the attack text when they design the detector at Line 192?
>
> A9:  We thank the reviewer for pointing this out. This was a typo — the goal of the detector is to distinguish between watermarked and non-watermarked text. We will revise this part accordingly.
>
> >Q10: Typo, figure and symbol unclear
>
> A10: We sincerely thank the reviewer for pointing it out. We will correct them in our revised manuscript.
>
> ---
> Anonymous Reference Link: https://docs.google.com/document/d/1t1HxJ5KkCydhf_AwUdqOC8R1c0bwY024Ryr_iTZo-2A/edit?usp=sharing

---

> > ### Comment · Reviewer_zCjb · 2025-04-06
> >
> > Thank the authors for the reply. I will maintain my score. I still think it's unreasonable for attackers to use much larger models. As for the adaptive watermarks, I was referring to cases where the watermark designers know this attack strategy and propose an adaptive defense against it.

---

> > > ### Author Response · Authors · 2025-04-07
> > >
> > > We are thankful to the reviewer **zCjb** for the time spent reviewing our paper. We address the concerns and questions below:
> > >
> > > > **Q1: Large model vs smaller attacker experiment**
> > >
> > > **A1:** We would like to respectfully point out that the setting where attackers are assumed to use larger models than the victim model has not been adopted in any prior work, to the best of our knowledge. If we adopt this setting in our orignal paper, we would hard to find suitable baseline methods to compare.
> > >
> > > We conduct experiment using Llama2-7b as generation model and apply Tiny(3b) as attack method.The experiment follows the same setting as our main experiments and we use 200 samples from the C4 dataset. Notably, the **baseline DIPPER and ChatGPT are larger than LLaMA2-7B**. Our findings are consistent with those in [1]: using a larger generation model does not necessarily make the watermark more robust. The results shown in below table, this further demonstrates the **strength of our attack** and questions the **necessity of assuming a stronger attacker**.
> > >
> > > | Attack      | KGW  | Uni  | UPV  | EWD  | DIP  | SIR  | EXP  |
> > > |-------------|------|------|------|------|------|------|------|
> > > | Del         | 23.1 | 1.8  | 6.2  | 20.8 | 56.4 | 42.8 | 9.6  |
> > > | Syn         | 84.8 | 17.2 | 65.4 | 75.4 | 99.8 | 82.0 | 52.6 |
> > > | GPT         | 98.8 | 62.6 | 72.4 | 90.4 | 99.6 | 60.2 | 73.6 |
> > > | DIPPER-2    | 94.4 | 45.8 | 60.2 | 89.0 | 99.6 | 62.6 | 81.2 |
> > > | SIRA-TINY   | 96.8 | 87.0 | 83.6 | 97.6 | 99.8 | 75.4 | 90.8 |
> > >
> > > > **Q2: Defencer know the attack strategy**
> > >
> > > **A2:** We would first like to emphasize that, to the best of our knowledge, **all existing text watermarking methods adopt high-entropy embedding strategies**. We would be very grateful if the reviewer could kindly provide a specific reference. **We will be more than happy to run the corresponding experiments and include the results in the revised version.**
> > >
> > > We now consider the case where the **defender is aware of our attack strategy**. The answer is that enforcing such a watermark in **low-entropy regions** of text would **significantly degrade generation quality** and may even **lead to hallucinations**. **Watermarking methods work by modifying the model’s output probabilities.** However, in low-entropy contexts, such manipulation of the logits can result in **unnatural token choices and thus compromise the output quality thus not feasible**. For example, given the prompt **"1 + 1 ="**, the token **"2"**  is overwhelmingly high probability and with low-entropy. Forcing the model to deviate from this most probable continuation—e.g., generating **"3"** or **"one"**—would yield **semantically or factually incorrect outputs**, harming both **fluency and accuracy**.
> > >
> > > Moreover, **low-entropy contexts inherently constrain token choice**, making it difficult to encode meaningful watermark signals **without being detectable by users** or **significantly increasing perplexity**. This not only weakens the effectiveness of watermarking but also leads to a **high false positive rate during detection**. A **rigorous theoretical treatment** of these challenges is stated in **KGW [2]**.
> > >
> > >
> > >
> > >
> > >
> > > **We thankful for the reviewer further feedback, and would sincerely appreciate it if you could consider increasing your score to reflect the revised manuscript**.
> > >
> > >
> > > ---
> > >
> > > [1] Liu, Aiwei, et al. *"An unforgeable publicly verifiable watermark for large language models."* ICLR (2023).
> > >
> > > [2] Kirchenbauer, John, et al. *"A watermark for large language models."* ICML (2023)

---

### Official Review · Reviewer_RVPR · 2025-03-16

**Overall Recommendation:** 3

**Summary:**

The paper introduces SIRA, a novel text watermark attack method that leverages the concept of self-information to efficiently and effectively remove watermarks from text generated by large language models. The authors conduct systematic experiments to demonstrate the effectiveness of their approach.

**Claims And Evidence:**

The paper highlights the vulnerability of watermarks placed on high self-information tokens, which forms the basis for SIRA as a targeted attack method. This claim is well supported by the results presented in Table 3. The authors also argue that self-information is better than entropy for identifying watermarked tokens, which is backed by the experiments in Appendix G, Table 9. Another claim is that even when the distribution of the attack model differs from the generation-time distribution, SIRA can still identify watermark locations, and larger attack models better estimate the generation distribution, leading to improved watermark removal. This claim is supported by the results in Figure 2.

However, there are a few claims that lack strong support. For instance, the authors state that even their most lightweight SIRA-Tiny method outperforms all previous approaches, but this comparison is tricky due to the varying attack strengths and inconsistent rankings across different metrics (e.g., Appendix F shows that on the Unigram watermark, SIRA-Large achieves better rewrite quality than DIPPER2, while Figure 3b shows that SIRA-Large has slightly worse quality than DIPPER2 on the same watermark). Another claim that lacks clear evidence is the $0.88 per million tokens cost, which is highlighted in the abstract but without a clear calculation provided.

There is also a claim in Appendix G, Table 9, that compares filtering potential green tokens using self-information, entropy, and probability.
I fully understand the self-information vs entropy part. The authors also attempt to prove that self-information is more accurate than probability, but I didn't follow this part. Since self-information and probability have a monotonic relationship, using percentiles should yield the same results. Without the code to reproduce the experiments and find the details, it is difficult to understand this discrepancy. The authors should clarify this point and potentially consider the possibility of numerical instability in their experiments.

**Essential References Not Discussed:**

I am not aware of any.

**Experimental Designs Or Analyses:**

Some of the claims in the paper are well-supported by the experimental designs and analyses, while others have weaker support. Please refer to the "Claims And Evidence" section for a detailed discussion.

**Methods And Evaluation Criteria:**

The methods and evaluation criteria used in the paper seems to be sound and appropriate.

**Other Comments Or Suggestions:**

I have no additional comments.

**Other Strengths And Weaknesses:**

The paper presents a novel method and conducts systematic experiments to support its claims. Some claims are well-supported, while others have weaker support. Please refer to the "Claims And Evidence" section for a detailed discussion.

**Questions For Authors:**

I noticed that the current s-BERT measurement compares the similarity to the non-watermarked text, which leads to the observation that the s-BERT score after paraphrasing attacks is even higher than the no-attack case. Also in Table 2, with more attacks, s-BERT score is higher. This is because strong paraphrasing models are used, making the score less informative about the degree of semantic preservation after the attack and more about the ability of the paraphrasing model. Why not measure the similarity with respect to the watermarked text instead?

**Relation To Broader Scientific Literature:**

The paper cites and compares its approach to many relevant baseline methods.

**Theoretical Claims:**

Although Appendix H contains some bounds, they are not strongly connected to the main text. The paper's main contribution lies in its practical aspects rather than its theoretical claims.

---

> ### Author Rebuttal · Authors · 2025-03-31
>
> We are thankful to the reviewer **RVPR** for the valuable time and effort spent reviewing our paper. We elaborate on the questions raised by the reviewer below:
>
> > **Q1: Clarification regarding claims: SIRA-Tiny method outperforms all previous approaches**
>
> **A1:** We would like to clarify that this statement only refers to the fact that SIRA-Tiny outperforms previous methods in terms of ASR under the same watermark strength. We agree that a more precise statement would be better. We will revise the text to reflect this more precise phrasing to avoid overclaim and add a corresponding note in Appendix C, Table 6.
>
> > **Q2: Lack of support: Claim SIRA-Large achieves better quality than DIPPER2, while Figure 3b shows that SIRA-Large has slightly worse quality than DIPPER2 on the same watermark**
>
> **A2:** We would like to clarify that in watermarking, text quality is typically referred to the perplexity metric, which is shown in Figure 3a. In contrast, Figure 3b, which the reviewer referenced, reports S-BERT scores that evaluate semantic preservation, a different aspect from text quality. Therefore, we respectfully argue that our claim that SIRA-Large achieves better quality than DIPPER2 remains valid. We appreciate the reviewer's feedback and will revise the related part more clearly to avoid potential misunderstandings.
>
> > **Q3: Computation regarding the cost**
>
> **A3:** We provide a more comprehensive cost analysis here and will include it in the relevant section of the manuscript. Specifically, we estimate the cost of processing 1M tokens of watermarked text using third-party services. According to OpenAI’s pricing, using the GPT Paraphraser (GPT-4o) would cost \$10 × 2 = \$20 (input + output). In contrast, our method (SIRA-Small) costs \$0.22 × 2 (input + output) × 2 (two iterations) = \$0.88, based on the AWS Bedrock LLaMA3-8b pricing. And the cost could be further reduced if we use SIRA-Tiny (LLaMA3-3b).
>
> > **Q4: Why is the self-information result different from probability?**
>
> **A4:** We clarify that the self-information in our method is conditional and chunk-based (e.g., calculated and percentile by segment), whereas the raw token probabilities are derived from the LLM’s output logits. This means that the statistical basis for the two measures is not aligned: the conditioning context and granularity differ. As a result, percentile ranks based on self-information may diverge from those based on raw probability. This mismatch leads to divergent percentile rankings and explains why filtering using self-information yields different token selections than filtering using raw probability. We will release our code to ensure reproducibility and transparency, and we appreciate the suggestion regarding potential numerical differences, as we will further increase the sample size to reduce numerical instability.
>
> > **Q5: Why not measure the similarity with respect to the watermarked text instead?**
>
> **A5:** **For the attack cases shown in Figure 3b and Table 2, our SBERT-based similarity is indeed computed between the attack text and the watermarked text as proposed by the reviewer**. As SBERT needs one pair of texts, "No Attack" is a special case; comparing the watermarked text to itself would be meaningless. Therefore, for “No Attack” we compare the watermarked text with the non-watermarked text. This term aims to reflect how the watermark algorithm changed the semantics. We acknowledge that this was not clearly explained in the paper. We will clarify this to avoid potential misunderstanding.

---

### Decision · Program_Chairs · 2025-05-01

**Decision:**

Accept (poster)

**Comment:**

This paper studies text watermarking for large language models and presents SIRA, a novel attack method designed to remove text watermarks from outputs of LLMS. As watermarking algorithms tend to embed signals in tokens with high self-information, SIRA identifies these high-entropy tokens, masks them, and then reconstructs the sequence using a paraphrased version of the original output. This targeted paraphrasing strategy aims to preserve semantic fidelity while disrupting the watermark signal.

Overall, the reviewers found the paper has several merits, including novel use of self-information, effective paraphrasing strategy, strong experimental results. Some of the reviewers raised concerns about the trade-off between semantic preservation and effectiveness of the attack, and lack of comparisons in some aspects.

Therefore, based on the reviews, author responses, and discussions among AC and reviewers, the paper is marginally above borderline accept. I would recommend Weak Accept.